EMBO
Molecular Medicine

# Role of bulge epidermal stem cells and TSLP signaling in psoriasis

Nuria Gago-Lopez[1], Liliana F Mellor[1], Diego Megías[2], Guillermo Martín-Serrano[3], Ander Izeta[4], Francisco Jimenez[5] & Erwin F Wagner[1,6,*] iD

## Abstract

Psoriasis is a common inflammatory skin disease involving a cross-talk between epidermal and immune cells. The role of specific epidermal stem cell populations, including hair follicle stem cells (HF-SCs) in psoriasis is not well defined. Here, we show reduced expression of c-JUN and JUNB in bulge HF-SCs in patients with scalp psoriasis. Using lineage tracing in mouse models of skin inflammation with inducible deletion of c-Jun and JunB, we found that mutant bulge HF-SCs initiate epidermal hyperplasia and skin inflammation. Mechanistically, thymic stromal lymphopoietin (TSLP) was identified in mutant cells as a paracrine factor stimulating proliferation of neighboring non-mutant epidermal cells, while mutant inter-follicular epidermal (IFE) cells are lost over time. Blocking TSLP in psoriasis-like mice reduced skin inflammation and decreased epidermal proliferation, VEGFα expression, and STAT5 activation. These findings unravel distinct roles of HF-SCs and IFE cells in inflammatory skin disease and provide novel mechanistic insights into epidermal cell interactions in inflammation.

**Keywords** epidermal hyper-proliferation; hair follicle stem cells; lineage tracing; psoriasis; thymic stromal lymphopoietin
**Subject Categories** Stem Cells & Regenerative Medicine; Skin

## Introduction

The epidermis is a stratified epithelium undergoing constant renewal due to the presence of stem cells (Alberts *et al*, 2002). It is composed of the inter-follicular epidermis (IFE), which consists of layers of keratinocytes, hair follicles, and associated sebaceous and sweat glands (Fuchs, 2008). Within the IFE, epidermal stem cells are localized in the basal cell layer. Upon commitment to terminal differentiation, basal stem cells asymmetrically divide giving rise to post-mitotic suprabasal layers establishing an epidermal barrier (Fuchs, 2008). In addition to the IFE, distinct regions in the hair follicle contain hair follicle stem cells (HF-SCs), such as the bulge, isthmus and junctional zone of hair follicle, and sebaceous glands (Liu *et al*, 2003; Horsley *et al*, 2006; Jensen *et al*, 2009). Differential expression of specific markers characterizes the different stem cell subpopulations, allowing the study of their specific contributions to different aspects of cell behavior. For instance, in mice, HF-SCs in the bulge region express CD34, while both HF-SCs and IFE basal keratinocytes express integrin-α6 (CD49f; Liu *et al*, 2003; Trempus *et al*, 2003; Horsley *et al*, 2006, 2008; Jaks *et al*, 2008; Jensen *et al*, 2009; Snippert *et al*, 2010). These different stem cell populations from IFE and hair follicles also have distinct roles during skin remodeling, wound healing, and homeostasis. For example, during homeostasis, committed progenitor cells (transit-amplifying keratinocytes) of the IFE maintain the tissue renewal, while HF-SCs remain largely quiescent until the onset of HF growth (Ito *et al*, 2005; Blanpain & Simons, 2013). During wound healing and acute inflammation, both IFE epidermal stem cells and K15[+] bulge HF-SCs contribute to early wound re-epithelialization by increasing their proliferation rates, thereby expanding the pool of progenitor cells that migrate out of the follicles toward the wound (Ito *et al*, 2005; Levy *et al*, 2005; Arwert *et al*, 2012; Mascre *et al*, 2012; Blanpain & Simons, 2013). Furthermore, it is reported that basal and squamous skin cell carcinomas arise from K15[+] bulge HF-SCs (Trempus *et al*, 2007; Lapouge *et al*, 2011; Schober & Fuchs, 2011; Wang *et al*, 2011). However, little is known about the specific contribution of K15[+] bulge HF-SCs and IFE stem cells to the development of chronic inflammatory skin diseases such as psoriasis.

Psoriasis is a common heterogeneous inflammatory disease of unknown etiology, with an estimated worldwide prevalence of 3% in the population (Wagner *et al*, 2010). The vast majority of patients develop a chronic "plaque-type" psoriasis referred to as psoriasis vulgaris. The most common sites of psoriasis are the elbows, knees, forearms, shins, retro-auricular regions, and scalp (Wolf-Henning

1   Genes, Development and Disease Group, Cancer Cell Biology Programme, Spanish National Cancer Research Centre (CNIO), Madrid, Spain
2   Confocal Unit at Spanish National Cancer Research Centre (CNIO), Madrid, Spain
3   Bioinformatics Unit at Spanish National Cancer Research Centre (CNIO), Madrid, Spain
4   Tissue Engineering Group, Biodonostia Health Research Institute, San Sebastian, Spain
5   Grupo de Patología Médica, Mediteknia Dermatologic Clinic, Universidad Fernando Pessoa Canarias, Universidad Las Palmas Gran Canaria, Las Palmas de Gran Canaria, Spain
6   Department of Dermatology and Department of Laboratory Medicine, Medical University of Vienna, Vienna, Austria
    *Corresponding author. Tel: +43 1 40400 78760; E-mail: erwin.wagner@meduniwien.ac.at

Boehncke, 2015). Scalp psoriasis is the most common type of psoriasis, representing 50% of the cases of psoriatic patients (Icen et al, 2009). Approximately 75–90% of patients with plaque-type psoriasis in other areas also develop psoriasis in the scalp (Ortonne et al, 2009). Psoriatic plaques are characterized and perpetuated by hyper-proliferation of keratinocytes, impaired differentiation and permanent infiltration of neutrophils and T cells (Wagner et al, 2010). Many studies have focused on immunological aspects of the disease; however, treatments against immunological targets are not completely effective, and recurrence and chronicity prevail. In an epidermal context, it has been demonstrated that human transit-amplifying keratinocytes in psoriatic plaques exhibit increased proliferation and a reduced apoptotic rate, fostering epidermal hyper-proliferation and altered differentiation in situ and in vitro (McKay & Leigh, 1995; Truzzi et al, 2011). However, it remains unclear which epidermal lineages give rise to psoriasis, since a longitudinal analysis of the behavior of psoriatic keratinocytes in human patients is not achievable.

Genetically engineered mouse models (GEMMs) of psoriasis have been generated to mimic various aspects of the disease (Wagner et al, 2010). We have previously generated an inducible epidermal-specific double knockout mouse model by epidermal deletion of c-Jun and JunB using the K5 promoter (referred as DKO*). This model develops a psoriasis-like disease within 2 weeks after induction, exhibiting several psoriatic hallmarks including hyper- and paraker-atosis, inflammatory infiltrate, elevated levels of cytokines/chemokines, increased subepidermal vascularization, and some co-morbidities like bone loss and arthritic joints (Zenz et al, 2005). Our goal in this study is to elucidate the specific contribution of different epidermal stem cell populations during the development of psoriasis-like disease, by applying a novel strategy to simultaneously track the IFE and K15$^+$ bulge HF-SCs stem cell lineages in the DKO*-psoriasis-like mouse model. This model allows us to permanently label different cell populations involved in the development of psoriasis-like disease in vivo, and to investigate the specific role of each population during disease initiation and development. Using this new mouse model, we found that a small fraction of K15$^+$ bulge HF-SCs can initiate a psoriasis-like disease. Mutant bulge HF-SCs are able to survive, whereas IFE cells disappear during disease progression. Furthermore, mutant bulge HF-SCs and basal keratinocytes secrete thymic stromal lymphopoietin (TSLP), which induces proliferation of neighboring non-mutant keratinocytes. These results emphasize the heterogeneity of epidermal stem cells and their role in psoriasis-like development, and define a mechanistic basis for epidermal hyperplasia through TSLP paracrine signaling.

# Results

## c-JUN and JUNB levels are reduced in bulge hair follicle stem cells (HF-SCs) from scalp psoriasis patients

Previous results have reported alterations in the expression levels of c-JUN and JUNB in the inter-follicular epidermis (IFE) of psoriatic plaques from human patients (Zenz et al, 2005); however, it is not known whether the expression levels of c-JUN and JUNB are affected in epidermal stem cells. While there are no specific markers to identify IFE stem cells, keratin 15 (K15)-positive cells represent

the vast majority of HF-SCs located in the bulge of hair follicles (Liu et al, 2003; Purba et al, 2014). Another marker that identifies putative bulge HF-SCs in human HFs is CD200 (Purba et al, 2014). We analyzed the expression of c-JUN and JUNB in the bulge region of HFs from scalp psoriatic patients, a common form of human psoriasis (Figs 1 and EV1A). K15 was highly expressed in hair follicles, and low expression was also found in some basal keratinocytes of the scalp epidermis (Fig 1A and B). K15$^+$ basal cells located in the outer root sheath (ORS) of the bulge region showed significantly reduced expression of c-JUN and JUNB in lesional hair follicles in anagen phase (38 and 12%, respectively), while c-JUN and JUNB were expressed in 70 and 50%, respectively, in non-lesional hair follicles and healthy scalp (Fig 1A–J). The reduction was confirmed in putative bulge HF-SCs identified by the surface marker CD200 (Fig EV1A).

Only bulge HF-SCs exhibited reduced c-JUN/JUNB expression, while basal and suprabasal ORS layers in the sub-bulge and proximal bulb regions maintained the expression of c-JUN/JUNB (Fig EV1B and C). In addition, other regulators of HF-SC renewal from other regions in the hair follicle, such as GATA-6, expressed in progenitor matrix cells (bulb) in mice were analyzed (Wang et al, 2017). Interestingly, GATA-6 was up-regulated in suprabasal ORS layers in the bulge region from psoriatic lesional scalps, while no differences were observed in GATA-6-positive cells from the proximal bulb (Fig EV1C). These findings show that scalp psoriasis is associated with a reduction of c-JUN and JUNB in bulge HF-SCs along with GATA-6 increased expression.

## Epidermal stem cell lineage tracing in the DKO* psoriasis-like mouse model

The differential expression of c-JUN and JUNB in CD200$^+$/K15$^+$ bulge HF-SCs in human samples of psoriatic plaques directed us to explore the potential causal contribution of epidermal stem cell populations to the initiation and development of psoriasis. To this end, we applied mouse genetics to ablate the expression of c-Jun and JunB, first globally in all epidermal stem cell populations present in both the basal layer of the IFE and hair follicle units during homeostasis (Byrne et al, 1994). We crossed the well-established DKO* psoriasis-like mouse model (Zenz et al, 2005) with the mT/mG fluorescent reporter mouse. The psoriasis-like phenotype in DKO*-mT/mG mice was induced by the genetic deletion of c-Jun and JunB floxed alleles in K5$^+$ basal keratinocytes and K5$^+$ epidermal stem cells after tamoxifen administration. This treatment also induced an irreversible replacement of the constitutive Tomato expression (red) by the expression of GFP only in K5$^+$ basal keratinocytes and K5$^+$ epidermal stem cells (Fig 2A), allowing tracing of epidermal stem cells and their progeny (transient-amplifying and differentiated keratinocytes) during psoriasis progression. Mice expressing Cre recombinase under the control of the K5 promoter with wild-type or heterozygous c-Jun and JunB were used as controls (Co-mT/mG).

We first analyzed the labeling efficiency and epidermal specificity of GFP in Co-mT/mG ear skin 15 days after tamoxifen injection. Without tamoxifen, 95 ± 5% of epidermal cells expressed Tomato and only few keratinocytes expressed GFP (Fig EV2A). Following tamoxifen treatment, Tomato expression decreased and GFP expression increased by ~ 60–70% in K5$^+$ epidermal cells of

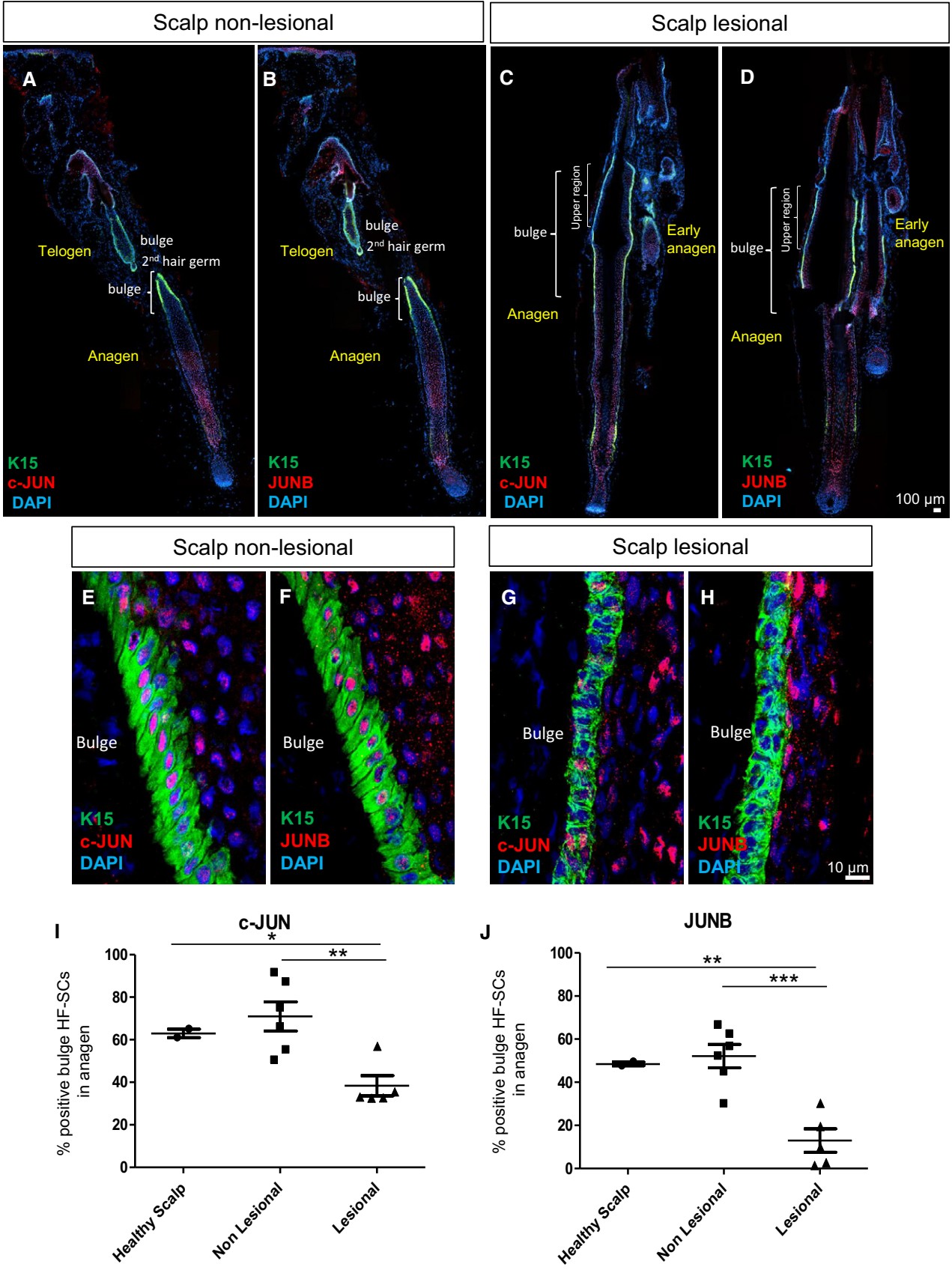

**Figure 1.**

◄

**Figure 1. Scalp psoriasis exhibits reduced expression of c-JUN and JUNB in hair follicle stem cells (HF-SCs).**

A–D  Representative composite immunofluorescence images of whole hair follicle units from non-lesional and lesional scalp psoriasis patients. K15 (green), c-JUN (red in A, C) and JUNB (red in B, D) and DAPI (blue).

E–H  Confocal images of the bulge region of human psoriatic hair follicles from non-lesional and lesional regions of the scalp. K15 (green), c-JUN (red in E, G) and JUNB (red in F, H) and DAPI (blue).

I, J  Percentage of HF-SCs (K15$^+$) that express c-JUN (I) and JUNB (J) in the bulge region of human psoriatic hair follicles from non-lesional and lesional scalp in comparison with healthy scalp. $n$ = 2–6 hair follicles in anagen per group from five psoriatic patients and two healthy patients. Data represent mean ± SD. Statistical significance *$P$ < 0.05, **$P$ < 0.01, ***$P$ < 0.001 (Student's two-tailed $t$-test relative to controls). See Appendix Table S2 for exact $P$-values.

Source data are available online for this figure.

IFE, hair follicles, and sebaceous glands (Fig EV2B, upper panels). DKO*-mT/mG mice treated with tamoxifen developed a psoriasis-like phenotype 2 weeks after induction, mainly in the ears, head, paws, and tail, as previously reported (Zenz *et al*, 2005). We next analyzed the expression of GFP in this psoriasis-like DKO*-mT/mG ear skin. Interestingly, although hyperplasia was present throughout the epidermis, only small patches of hyperplastic keratinocytes expressed GFP in IFE, hair follicles, and sebaceous glands, showing a mosaic pattern of mutant$^{GFP}$-positive keratinocytes among a majority of non-mutant$^{Tom}$-positive keratinocytes (Fig EV2B, lower panels). We confirmed that the deletion of c-Jun and JunB occurred only in mutant$^{GFP}$ epidermal cells using fluorescence microscopy (Fig EV2C). Therefore, our tracing system successfully allows studying the dynamics of mutant$^{GFP}$ epidermal stem cells and their progenitor cells, and non-mutant$^{Tom}$ epidermal stem cells and their progenitor cells in this psoriasis-like mouse model.

## Mutant and non-mutant IFE keratinocytes exhibit different proliferation and apoptosis rates during psoriasis-like development

To further understand the mosaic deletion pattern in the DKO*-mT/mG psoriasis-like mouse model, we traced GFP expression after tamoxifen injection during the initiation (day 0–5), mid-term (day 7–9), and late-term (day 15–30) stages of the disease (Fig 2B). Initial epidermal cells labeled for GFP$^+$ at day 0 after the last injection with tamoxifen are observed in isolated groups of cells in IFE and hair follicles (Figs 2C and EV2D). Composite images of whole ear sections showed that mutant$^{GFP}$ keratinocytes are increased at 5–7 days followed by a drastic reduction at days 15–30, whereas immune cell infiltration and epidermal thickening sharply increased over time (Fig 2C and D; Appendix Fig S1A–C).

We then analyzed apoptosis and proliferation by cleaved caspase-3 (cCas3) and Ki67 expression, respectively. Apoptosis was observed in mutant$^{GFP}$ and non-mutant$^{Tom}$ keratinocytes from the IFE of DKO*-mT/mG mice during psoriasis-like progression, although at day 7 after induction, the apoptosis rate was higher in mutant$^{GFP}$ keratinocytes (38%), when compared to non-mutant$^{Tom}$ keratinocytes (10%) (Fig 2E, upper panel). Interestingly, 78% of non-mutant$^{Tom}$ keratinocytes expressed Ki67 vs. 40% of mutant$^{GFP}$ keratinocytes, suggesting a proliferative advantage of non-mutant$^{Tom}$ keratinocytes (Fig 2E, lower panel). The rapid loss of mutant$^{GFP}$ keratinocytes in the ear skin of DKO*-mT/mG mice was also confirmed by intravital confocal imaging analyses at day 7 after induction (Fig 2F and G, and Movie EV1). Overall, these findings suggest that the psoriasis-like phenotype in the IFE of DKO*-mT/mG mice is characterized by different proliferation and apoptotic rates of

distinct mutant and non-mutant populations during psoriasis-like progression.

The global proliferation/apoptosis imbalance led to almost complete depletion of mutant$^{GFP}$ basal and suprabasal keratinocytes in the IFE at the latest stage (day 15–30). This drastic loss could imply that the psoriasis-like phenotype should also decrease over time. However, the psoriasis-like phenotype is sustained until the late term of disease progression (D30) (Fig 2D). Interestingly, remaining mutant$^{GFP}$ keratinocytes at the late term of disease progression were located in the outer root sheath (ORS) in both hair follicles and in adjacent IFE keratinocytes (Fig 2H and I). In fact, same dynamic pattern of GFP/Tomato cells was observed in the IFE of tail and back skin during psoriasis development in DKO* mice (Appendix Fig S2). Interestingly, mutant GFP$^+$ hair follicles from tail psoriatic skin underwent anagen induction suggesting the activation of these mutant$^{GFP}$ HF-SCs in psoriasis-like disease. Our next goal was to further investigate the contribution of these HF-associated mutant$^{GFP}$ keratinocytes during psoriasis-like progression.

## Deletion of c-Jun/JunB in bulge HF-SCs is sufficient for the development of inflammatory skin disease

Bulge HF-SCs are an important stem cell population for wound healing and for the initiation and maintenance of skin carcinomas, and are characterized by the expression of K15 and the surface marker CD34 in mice. We hypothesized that the remaining mutant$^{GFP}$ keratinocytes observed in the ORS of the hair follicles and in adjacent IFE regions are mainly derived from mutant HF-SCs. To test this hypothesis, we first conducted FACS analyses to determine the proportion of the different HF-SC subpopulations expressing CD34, CD49f, and Sca-1 markers (Jensen *et al*, 2010) in the ear skin of the Co-mT/mG and DKO*-mT/mG mice during the psoriasis-like progression (Appendix Fig S3A). As expected, the number of mutant$^{GFP}$ basal IFE cells decreased during disease progression, whereas non-mutant$^{Tom}$ basal and suprabasal IFE cells increased over time (Appendix Fig S3B and C). With regard to the hair follicles, the number of mutant$^{GFP}$ HF-SCs from bulge and junctional zone was increased twofold during disease progression, whereas the number of non-mutant$^{Tom}$ HF-SCs decreased over time (Appendix Fig S3D–F). To confirm that CD34$^+$ mutant$^{GFP}$ cells belong to bulge HF-SCs and not to the IFE epidermis, we conducted histological section and ear whole-mount confocal analyses and observed co-expression of GFP$^+$ and CD34$^+$ cells only in the bulge region of hair follicles (Fig EV3A and Appendix Fig S4A). Interestingly, both bulge HF-SC populations (mutant$^{GFP}$ and non-mutant$^{Tom}$) had reduced mRNA levels of the quiescence transcription factors Foxc1 and Nfatc1 (Horsley *et al*, 2008; Wang *et al*, 2016; Fig EV3B

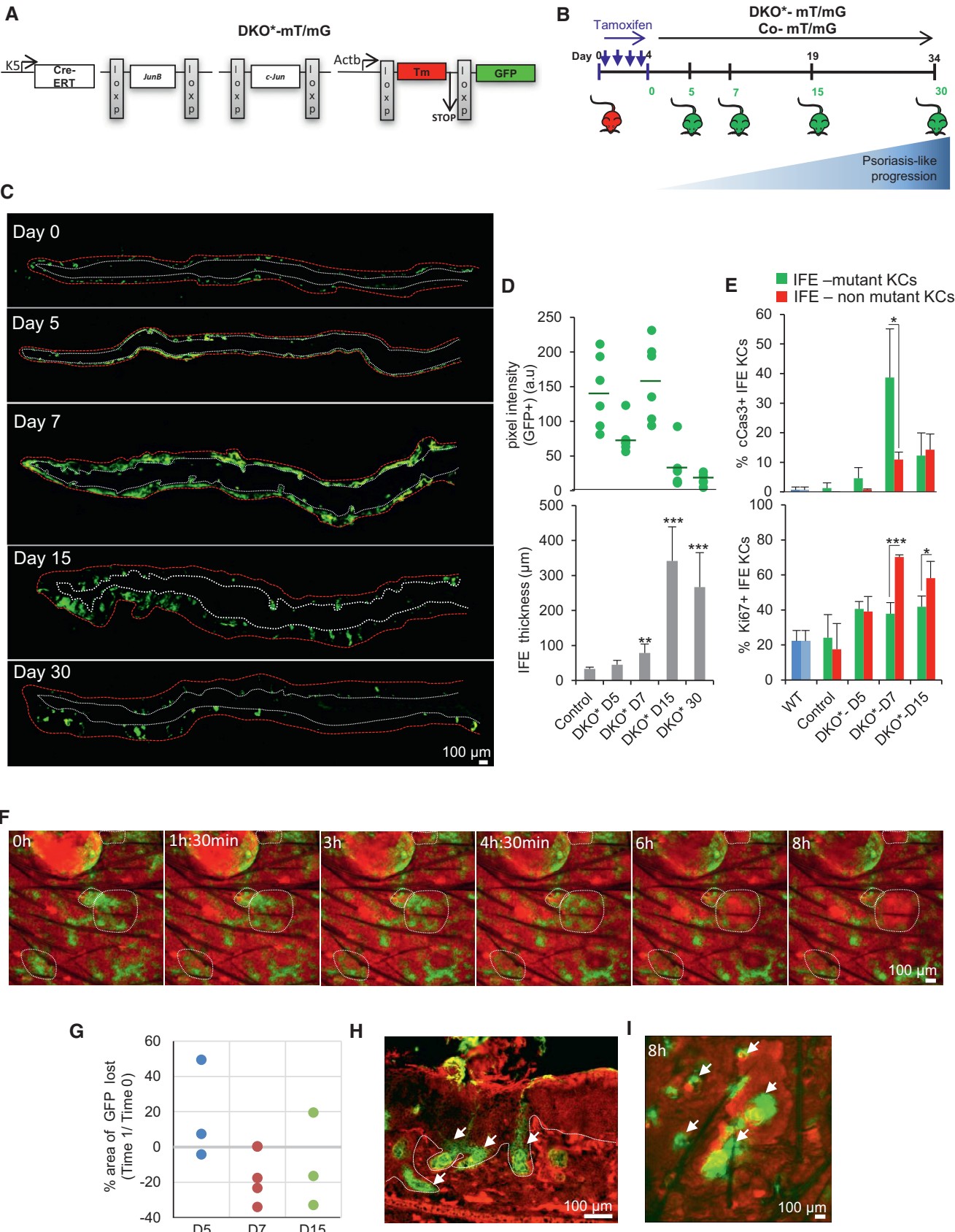

Figure 2.

**Figure 2.  Mutant and non-mutant inter-follicular keratinocytes have different proliferation and apoptotic rates during psoriasis-like disease progression.**

A   Schematic representation of the JunB and c-Jun double knockout mouse model (DKO*) with lineage tracing used to investigate the psoriasis-like disease development in mice. Briefly, DKO* psoriasis-like mouse model was generated by the cross of mice carrying the floxed JunB allele ($JunB^{f/f}$) and floxed c-Jun allele ($c$-$Jun^{f/f}$) with the transgenic mice expressing the Cre recombinase–estrogen receptor fusion under the control of the basal keratinocyte-specific K5 promoter (K5-Cre-ERT) to obtain $JunB^{f/f}$ $c$-$Jun^{f/f}$ K5-Cre-ERT mice. To trace epidermal-specific deletion of $JunB^{f/f}$ and $c$-$Jun^{f/f}$ in DKO* mice, global double-fluorescent Cre reporter mouse Gt(ROSA) 26Sor$^{tm4(ACTB-tdTomato,-EGFP)Luo/J}$ (referred as mT/mG) was crossed with DKO* mice (referred as DKO*-mT/mG).

B   Experimental timeline to induce psoriasis-like disease in 8-week-old mice with four consecutive doses of tamoxifen injections (2 mg). Upon induction, non-mutant keratinocytes express Tomato, while mutant keratinocytes express GFP. GFP expression was analyzed at 0, 5, 7, 15, and 30 days post-induction.

C   Composite immunofluorescence images showing GFP expression of whole ear sections from DKO*-mT/mG mice at day 5, 7, 15, and 30. Increased GFP expression is shown at day 7 followed by progressive decrease of GFP$^+$ keratinocytes (white dotted line represents basal layer, and red dotted line represents outermost skin layer). $n$ = 3 per time point.

D   Quantification analysis of GFP expression (top) and epidermal thickness (bottom) of DKO* mice at different time points during psoriasis-like disease progression. $n$ = 6 per time point. Data represent mean ± SD. Statistical significance **$P < 0.01$, ***$P < 0.001$ (Student's two-tailed $t$-test relative to control group). See Appendix Table S2 for exact $P$-values.

E   Quantification analysis of cleaved caspase-3 (cCas3, top) and Ki67 (bottom) in ear skin of DKO* mice at different time points during psoriasis-like disease progression. $n$ = 3 per time point. Data represent mean ± SD. Statistical significance *$P < 0.05$, ***$P < 0.001$ (Student's two-tailed $t$-test). See Appendix Table S2 for exact $P$-values.

F   Intravital confocal time-lapse imaging of non-mutant Tomato keratinocytes and mutant GFP keratinocytes of DKO*-mT/mG at day 7 after first tamoxifen injection. Dotted circles emphasize areas in which mutant GFP$^+$ keratinocytes are replaced by non-mutant Tomato$^+$ keratinocytes.

G   Percentage of GFP$^+$ area eliminated during intravital confocal time-lapse imaging at day 5, 7, and 15 in DKO*-mT/mG mice (Time-lapse 1:8 h). Positive value: Area of GFP increased. Negative value: Area of GFP reduced.

H   Fluorescence imaging of DKO*-mT/mG ear skin at day 30 after first tamoxifen injection shows that remaining mutant GFP$^+$ keratinocytes reside along the hair follicles. White dotted line separates epidermis and dermis. White arrows represent GFP$^+$ hair follicles.

I   Intravital confocal time-lapse imaging of DKO*-mT/mG at day 15 after first tamoxifen injection. Mutant GFP$^+$ keratinocytes (white arrows) are maintained around hair follicles. White arrows represent GFP$^+$ hair follicles.

Source data are available online for this figure.

and C), suggesting an activation of bulge HF-SCs during psoriasis progression. In fact, FACS-isolated CD34$^+$ mutant$^{GFP}$ bulge HF-SCs displayed an increased clonogenic capacity *in vitro*, compared to non-mutant$^{Tom}$ bulge HF-SC counterparts and control bulge HF-SCs (Fig EV3D and Appendix Fig S4B). Overall, all these findings suggest that bulge HF-SCs are activated and exit quiescence in the psoriasis-like disease.

We then investigated whether mutant$^{GFP}$ keratinocytes from bulge HF-SCs are sufficient to induce and maintain skin inflammation. We generated a new DKO* model to specifically target K15$^+$ HF-SCs and termed it DKO*$^{K15}$-mT/mG. This was achieved by crossing mice carrying the floxed JunB ($JunB^{f/f}$) and c-Jun ($c$-$Jun^{f/f}$) alleles with a transgenic mouse line expressing the Cre recombinase–prostaglandin receptor fusion under the control of keratin 15 promoter (K15-Cre-PGR; Liu *et al*, 2003). After mifepristone (RU-486) treatment, this model allowed the specific targeting of K15$^+$ HF-SCs in the bulge to delete c-Jun and JunB and to render targeted cells to become GFP positive (Fig 3A). Interestingly, DKO*$^{K15}$-mT/ mG mice developed a similar psoriasis-like phenotype as the DKO* mouse model 2 weeks after induction, displaying similar histological and chronological features in the skin at day 30 of the disease (Fig 3B–E). However, the psoriasis-like severity was less pronounced in DKO*$^{K15}$ mice when compared to the DKO* mice. The expression of specific pro-inflammatory cytokines/alarmins, such as IL-17A, IL-6, and lipocalin-2 (LCN2), was higher in the serum of DKO* mice compared to DKO*$^{K15}$ mice at day 30 after psoriasis-like induction (Fig 3F–H). These cytokines and LCN2 were significantly increased in DKO*$^{K15}$ mice in comparison with control mice, suggesting a mild systemic inflammatory process in the HF-SC DKO*$^{K15}$ psoriasis-like mouse model. We next analyzed the dynamic GFP expression in the DKO*$^{K15}$-mT/mG ear skin and compared it to DKO*-mT/mG mice during psoriasis-like progression. At day 0 after the last injection with mifepristone, the majority of hair follicles are labeled for GFP including some isolated spots

found in the IFE (Fig EV3E). During initiation of the psoriasis-like disease (day 5), 20% of mutant$^{GFP}$ epidermal cells derived from mutant$^{GFP}$ K15$^+$ bulge HF-SCs were quantified by FACS analysis (CD45$^-$CD49f$^+$CD34$^-$ epidermal total population), whereas mutant$^{GFP}$ epidermal cells derived from mutant$^{GFP}$ K5$^+$ basal keratinocytes/HF-SCs was increased by 35–50% (Fig 3I). In contrast, during late stages of the psoriasis-like disease (day 15–30), the percentage of mutant$^{GFP}$ epidermal cells was only twofold reduced in DKO*$^{K15}$ mice (from 20 to 10%), whereas in DKO* mice, it was drastically reduced fivefold at days 15–30 (from 35–50 to 8%). Immunofluorescence analyses for c-Jun and JunB expression confirmed that the mutant$^{GFP}$ keratinocytes derived from mutant$^{GFP}$ K15$^+$ bulge HF-SCs did not express c-Jun or JunB (Fig EV3F). Interestingly, DKO*$^{K15}$-mT/mG psoriatic-like mice showed the same mosaic pattern of green and Tomato epidermal cells than the one previously described in the DKO* mouse model at the late term of psoriasis-like development (Fig 3J). In fact, a similar proliferation pattern of mutant$^{GFP}$ and non-mutant$^{Tom}$ keratinocytes was observed in both models, although the apoptosis rate of mutant$^{GFP}$ keratinocytes was reduced in DKO*$^{K15}$-mT/mG mice when compared to DKO*-mT/mG mice (Figs 3K and 2E).

Overall, these data show that the specific inactivation of c-Jun and JunB in bulge HF-SCs is sufficient to induce a psoriasis-like phenotype in the DKO*$^{K15}$-mT/mG mice. Importantly, these data uncovered that bulge HF-SCs may act as a reservoir of new mutant epidermal cells, sustaining epidermal hyperplasia and skin inflammation, thereby recapitulating the observed skin phenotype of DKO* mice at later time points (Fig EV3G).

### Keratinocytes derived from mutant$^{GFP}$ HF-SCs have increased cell survival compared to mutant$^{GFP}$ keratinocytes derived from IFE

To determine whether keratinocytes derived from mutant$^{GFP}$ bulge HF-SCs behave differently than IFE mutant$^{GFP}$ basal keratinocytes

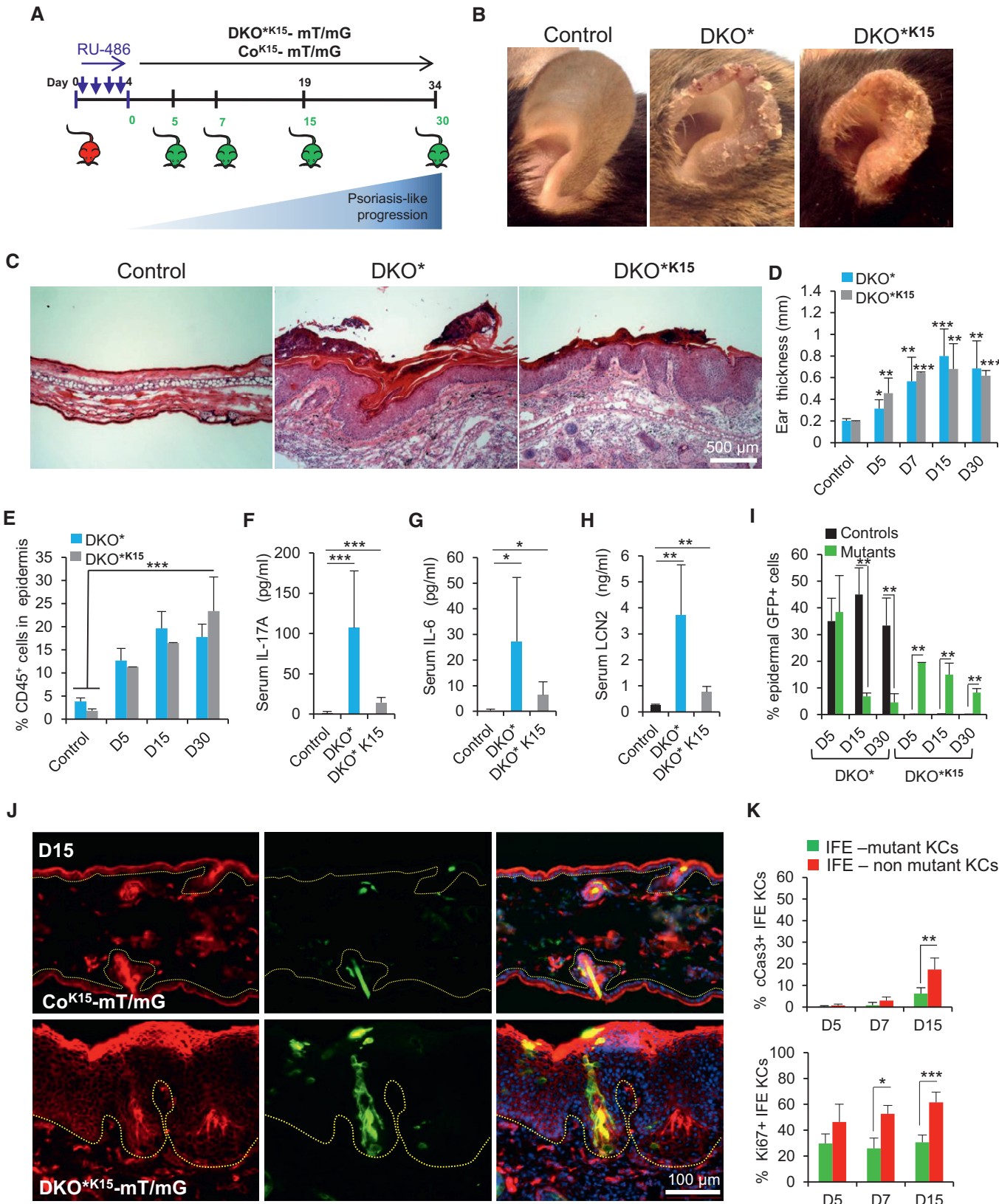

**Figure 3.**

◀

**Figure 3.  Inactivation of c-Jun and JunB in bulge HF-SCs is sufficient to initiate psoriasis-like development.**

A   Experimental timeline showing the different time points analyzed after mifespristone induction (day 5, 7, 15, and 30, green) to evaluate GFP expression in ear skin from control (Co[K15]-mT/mG) and mutant (DKO*[K15]-mT/mG) mice.

B   Ear pictures of control, DKO*, and DKO*[K15] mice after 15 days of induction show clear signs of inflammation and psoriatic-like scales in both models.

C   Representative images of hematoxylin and eosin (H&E) staining of ear sections from control, DKO*, and DKO*[K15] mice.

D   Ear thickness measurement at different time points during psoriasis-like development in control, DKO*, and DKO*[K15] mice. $n > 5$ per group. Data represent mean $\pm$ SD. Statistical significance *$P < 0.05$, **$P < 0.01$, ***$P < 0.001$ (Student's two-tailed $t$-test relative to control group). See Appendix Table S2 for exact $P$-values.

E   FACS quantification of CD45[+] cells from ear skin of control, DKO*, and DKO*[K15] mice at different time points during psoriasis-like development. $n > 5$ per group. Data represent mean $\pm$ SD. Statistical significance *$P < 0.05$, **$P < 0.01$, ***$P < 0.001$ (Student's two-tailed $t$-test relative to controls). See Appendix Table S2 for exact $P$-values.

F–H   Quantification of IL-17A, IL-6, and LCN2 production in sera of control, DKO*, and DKO*[K15] mice at day 30 of psoriasis-like disease by ELISA. $n > 4$ per group. Data represent mean $\pm$ SD. Statistical significance *$P < 0.05$, **$P < 0.01$, ***$P < 0.001$ (Student's two-tailed $t$-test relative to controls). See Appendix Table S2 for exact $P$-values.

I   Quantification of total number of GFP[+] epidermal cells by FACS analysis from DKO*-mT/mG and DKO*[K15]-mT/mG mice at different time points during psoriasis-like progression compared to Co-mT/mG mice. $n = 3$–6 per group and time point. Data represent mean $\pm$ SD. Statistical significance **$P < 0.01$ (Student's two-tailed $t$-test relative to control group). See Appendix Table S2 for exact $P$-values.

J   Representative immunofluorescence images of ear sections from Co[K15]-mT/mG and DKO*[K15]-mT/mG mouse models after 15 days of induction. Red (Tomato), green (GFP), and blue (DAPI). Hyperplasia of Tomato[+] keratinocytes in psoriatic-like DKO*[K15]-mT/mG mouse with clusters of GFP[+] keratinocytes is derived from hair follicle K15[+] GFP[+] stem cells. $n = 3$ per group. Dotted lines separate epidermis and dermis.

K   Quantification analysis of cleaved caspase-3 (cCas3) and Ki67 in ear skin of DKO*[K15]-mT/mG mice at different time points during psoriasis-like disease progression. $n = 3$–5 per group. Data represent mean $\pm$ SD. Statistical significance *$P < 0.05$, **$P < 0.01$, ***$P < 0.001$ (Student's two-tailed $t$-test). See Appendix Table S2 for exact $P$-values.

Source data are available online for this figure.

(b-KCs), we FACS-isolated and cultured these two populations at low concentration of 2,000 cells/cm², as well as GFP[+] bulge HF-SCs and b-KCs from Co-mT/mG mice as controls (Fig 4A). Next, we conducted a time-lapse confocal microscopy analysis of small colonies after 3 days in culture to evaluate the activity of different cell populations. Interestingly, over a 48-h time course, mutant[GFP] KCs derived from mutant[GFP] bulge HF-SCs showed similar growth compared to control[GFP] KCs from bulge HF-SCs (Fig 4B), while mutant[GFP] b-KCs had a rapid growth during the first 24 h followed by a drastic decrease in cell number due to cell death compared to control[GFP] b-KCs (Fig 4C and Movie EV2). These results show that the same mutation in primary keratinocytes from distinct epidermal stem cell populations induces different proliferation and survival phenotypes, and suggest that bulge HF-SCs play a major role in the maintenance of the psoriasis-like disease.

**Secreted factors from mutant[GFP] keratinocytes induce hyper-proliferation of neighboring non-mutant[Tom] keratinocytes**

We have shown that in both psoriasis-like mouse models, DKO*-mT/mG and DKO*[K15]-mT/mG, non-mutant[Tom] KCs mainly sustain epidermal hyperplasia. We hypothesized that the establishment of cell–cell interactions between mutant[GFP] KCs and non-mutant[Tom] KCs primed the non-mutant[Tom] population to increase their proliferation, leading to chronic hyperplasia in the psoriasis-like disease. To determine if the hyper-proliferation of non-mutant[Tom] KCs is conditioned by the presence of mutant[GFP] KCs, we co-cultured sorted mutant[GFP] bulge HF-SCs and mutant[GFP] b-KCs with their corresponding non-mutant[Tom] bulge HF-SC and b-KC counterparts (Fig 4D). Mutant[GFP] KCs derived from both bulge HF-SC and b-KC populations were able to stimulate the proliferation of co-cultured neighboring non-mutant[Tom] KCs (Fig 4E and F, and Movie EV3). This increase in the proliferation of Tomato[+] KCs was also observed when mutant[GFP] bulge HF-SCs and mutant[GFP] b-KCs were co-cultured with control[Tom] bulge HF-SC and b-KC counterparts (Appendix Fig S5A and B). In addition, non-mutant[Tom] bulge

HF-SCs conditioned the proliferation of mutant[GFP] bulge HF-SCs, whereas non-mutant[Tom] b-KCs did not affect the proliferation of mutant[GFP] b-KCs (Fig 4E and F). Interestingly, mutant[GFP] KCs from b-KCs also induced an increased cell death in neighboring non-mutant/control[Tom] KCs, while mutant[GFP] KCs from bulge HF-SCs did not induce a significant increase in cell death of their non-mutant/control[Tom] KC counterparts (Appendix Fig S5C and D). To determine whether the observed increased proliferation and cell death are a result of a direct cell–cell contact rather than stimulation by secreted factors, we cultured b-KCs in trans-wells, in which the non-mutant[Tom] b-KCs were cultured at the bottom of the plate and the mutant[GFP] b-KCs were cultured on the trans-well inserts. We observed similar cell proliferation and death rates as the ones observed in the co-cultures, suggesting that cell–cell contact is not required for the hyper-proliferation of non-mutant[Tom] KCs (Fig 4G). These observations suggest an induction of a proliferative advantage in non-mutant[Tom] KCs by paracrine factors secreted from mutant[GFP] KCs, independently of their origin from the hair follicle or IFE during the psoriasis-like progression in the DKO* mouse model.

**Mutant and non-mutant epidermal stem cells display different molecular signatures**

To gain insight into the potential molecular mechanisms that govern the behavior of the distinct mutant and non-mutant bulge HF-SCs and IFE cell populations during the progression of the psoriasis-like disease, we performed mRNA sequencing analyses. We FACS-isolated GFP[+], Tomato[+] bulge HF-SCs and b-KCs from DKO*-mT/mG mice, and their counterpart populations from control mice at the mid-term of the psoriasis-like progression (Fig EV4A). Interestingly, non-mutant[Tom] populations exhibited a higher differential gene expression (972 total genes in HF-SCs and 632 total genes in b-KCs) than the one observed in mutant[GFP] populations (701 total genes in HF-SCs and 444 total genes in b-KCs) (Fig 5A, upper panel). A Venn diagram including these differentially expressed genes (DEG) was generated to depict the results in each population

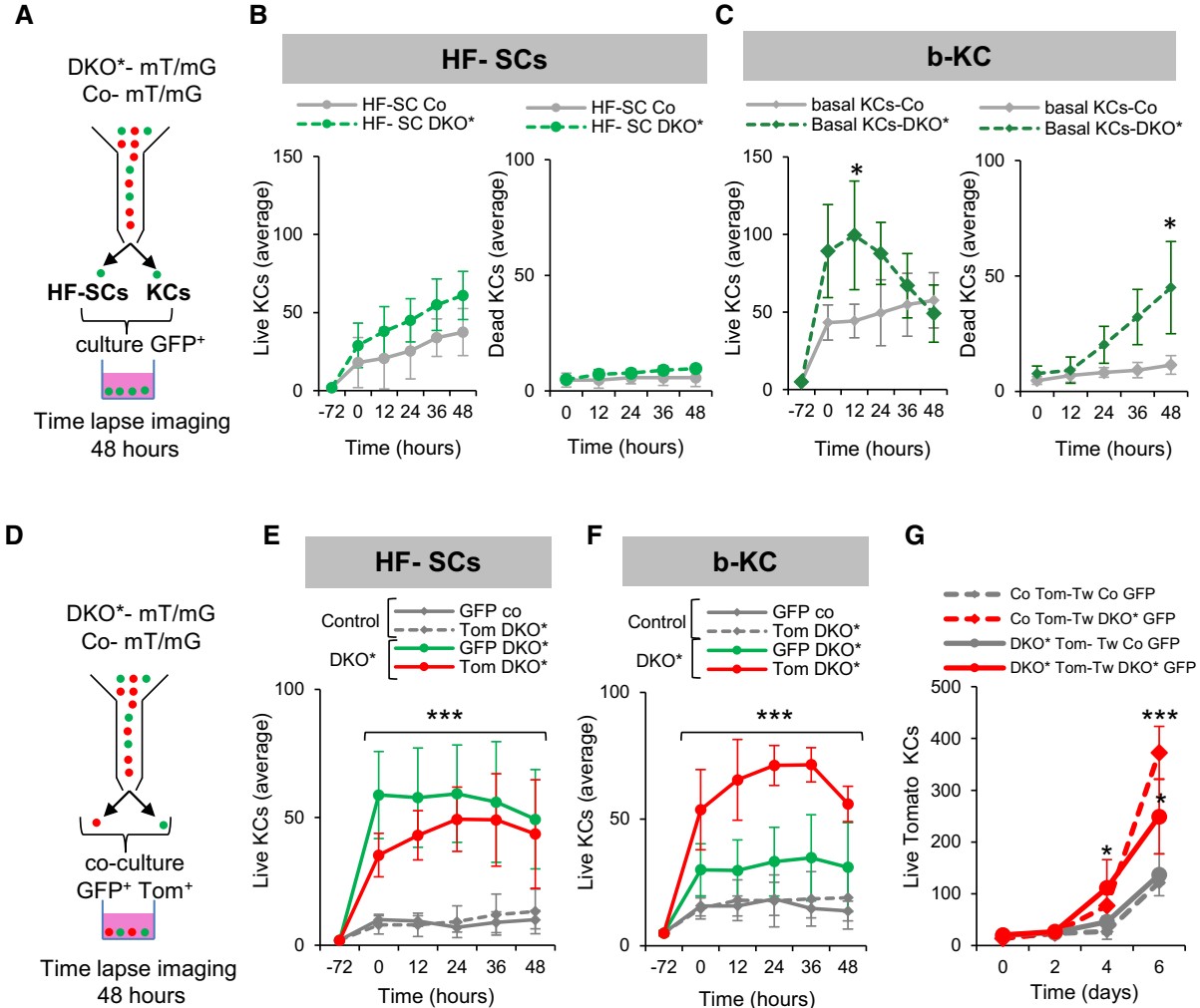

**Figure 4. Differential behavior of inter-follicular and follicular stem cells.**

A   Schematic representation of cell sorting and single culture conditions. GFP⁺ bulge hair follicle stem cells (HF-SCs, CD34⁺ CD49f^high) and GFP⁺ basal keratinocytes (b-KC, CD49f^high) from ear skin of Co-mT/mG and DKO*-mT/mG mice were collected at day 10 after tamoxifen induction, sorted by FACS, and cultured at low density (2,000 cells/cm²) during 3–5 days as primary cultures. Time-lapse imaging of six fields was performed during 48 h, once small colonies of 5–10 cells were observed.

B   Average number of live and dead (TOPRO-3⁺ DAPI⁺) primary keratinocytes derived from HF-SC^GFP at different time points of time-lapse capture during 48 h. There is a slight increase over time of live HF-SCs from DKO* mice when compared to control cells, and no difference in average of dead cells. *n* = 4 fields from two independent experiments. Data represent mean ± SD. Non-statistical significance (two-way ANOVA and Bonferroni post-test).

C   Average number of live and dead (TOPRO-3⁺ DAPI⁺) primary keratinocytes derived from basal keratinocytes^GFP show a significant increase in DKO* basal live keratinocytes at 12 h, followed by a significant increase in cell death at 48 h. *n* = 4 fields from two independent experiments. Data represent mean ± SD. Statistical significance *P < 0.05 (two-way ANOVA and Bonferroni post-test).

D   Schematic representation of cell sorting and co-culture conditions. GFP⁺ and Tom⁺ bulge HF-SCs (CD34⁺ CD49f^high) and GFP⁺ and Tom⁺ basal keratinocytes (CD49f^high) from ear skin of Co-mT/mG and DKO*-mT/mG collected at day 10 after tamoxifen induction were sorted by FACS and co-cultured (same proportion of GFP and Tom cells) at low density (2,000 cells/cm²) during 3–5 days. When small colonies of 5–10 cells were formed, time-lapse imaging of six fields was performed during 48 h.

E   Average number of live primary keratinocytes from co-cultures of GFP⁺ with Tomato⁺ bulge HF-SCs at different time points of time-lapse capture during 48 h. Co-culture of HF-SC^GFP with HF-SC^Tom from DKO*-mT/mG mice shows a significant increase of HF-SC^Tom, while the HF-SC^Tom from DKO*-mT/mG mice show a normal growth when they are co-cultured with HF-SC^GFP from control mice. *n* = 4 fields from two independent experiments. Data represent mean ± SD. Statistical significance ***P < 0.001 (two-way ANOVA and Bonferroni post-test).

F   Average number of live primary keratinocytes from co-cultures of GFP⁺ with Tomato⁺ basal keratinocytes at different time points of time-lapse capture during 48 h. Co-culture of b-KC^GFP with b-KC^Tom from DKO*-mT/mG mice shows significant increase of b-KC^Tom, while these b-KC^Tom from DKO*-mT/mG mice show a normal growth when they are co-cultured with b-KC^GFP from control mice. *n* = 4 fields from two independent experiments. Data represent mean ± SD. Statistical significance ***P < 0.001 (two-way ANOVA and Bonferroni post-test).

G   Average total number of live Tomato⁺ basal keratinocytes (from DKO* or control mice) at different time points cultured in the bottom part of trans-well system cultures with GFP⁺ basal keratinocytes (from DKO* or control mice) cultured in the upper trans-well (Tw). *n* = 5 fields from three independent experiments. Data represent mean ± SD. Tom⁺ basal keratinocytes significantly increased when these were cultured with mutant GFP⁺ basal keratinocytes in the trans-well in comparison with control GFP⁺ basal keratinocytes. Statistical significance *P < 0.05, ***P < 0.001 (two-way ANOVA and Bonferroni post-test).

Source data are available online for this figure.

(Fig EV4B and C). In fact, the bulge HF-SC population displayed more up-regulated genes, including genes specifically up-regulated in non-mutant[Tom] HF-SCs (> 400 genes), mutant[GFP] HF-SCs (> 150 genes) and genes commonly up-regulated in both populations of HF-SCs (GFP/TOM > 200 genes; Fig 5A). Conversely, the mutant b-KC population presents the majority of down-regulated genes (> 200 genes; Fig 5A), whereas around 100 common genes were up-regulated in all cell populations (Fig 5A).

Next, we conducted gene ontology (GO) analyses, to identify the specific enrichment of the obtained genetic signatures in different biological processes, focusing attention on the bulge HF-SCs up-/down-regulated genes and on the common genes up-regulated in all cell populations (Figs 5B, and EV4D and E). The most prominent biological process enriched in the genetic signatures of all cell populations are related to neutrophil degranulation and chemotaxis, chronic inflammatory responses, and epidermal cell differentiation (Fig EV4D). These findings support our prior findings observed *in vivo* and *in vitro*, where non-mutant[Tom] KCs acquire psoriasis-like hallmarks without having c-Jun and JunB mutation.

Mutant[GFP] bulge HF-SCs displayed a specific enrichment of genes encoding pro-inflammatory mediators in psoriasis, in particular an increase in genes related to arachidonic acid (AA) metabolism, such as Ptgs2 (Figs 5B and EV4F), and specific pro-inflammatory cytokines, such as Tnf-α, Il-23, and Il1-α (Fig EV4G–I), whereas genes related to epithelial cell migration were down-regulated (Fig EV4E). In contrast, non-mutant[Tom] bulge HF-SCs showed an enrichment of genes related to angiogenesis, VEGF receptors, extracellular matrix disassembly, and epithelial cell migration and proliferation (Fig 5B). Both mutant[GFP] and non-mutant[Tom] bulge HF-SCs shared up-regulated genes related to mitotic cell cycle and asymmetric cell division (Fig 5B). These results are in agreement with our previous observations that both bulge HF-SC populations were active during psoriasis progression. In addition, other pro-inflammatory cytokines important in psoriasis, such as Il-6, were only up-regulated in non-mutant[Tom] cell populations (Fig EV4J), and we also observed that Il-1β was significantly up-regulated in non-mutant[Tom] b-KCs (Fig EV4K). These data suggest that mutant[GFP] bulge HF-SCs have an important pro-inflammatory role in the initiation of psoriasis-like disease and unravel the expression of a unique transcriptomic profile of non-mutant[Tom] bulge HF-SCs in the development of psoriasis.

We next investigated the differential expression of genes coding for secreted proteins in mutant[GFP] populations of bulge HF-SCs and b-KCs, since our *in vitro* results showed that mutant[GFP] cell populations secreted paracrine factors that induced the proliferation of non-mutant[Tom] KCs. Four secreted proteins were overexpressed Tslp, Psors1c2, Fetub, and Lif (Fig 5C). All four proteins are involved in systemic inflammation and TSLP, PSORS1C2 and LIF in inflammatory skin diseases (Asumalahti *et al*, 2000; Szepietowski *et al*, 2001; He *et al*, 2008). Thymic stromal lymphopoietin (TSLP) is known for its important role in allergies and atopic dermatitis (He *et al*, 2008), and it is up-regulated in the epidermis of psoriatic patients (Volpe *et al*, 2014). We detected a 12-fold increase in mutant[GFP] bulge HF-SCs and a fivefold increase in mutant[GFP] b-KCs at the early/mid-term (D7) of the psoriasis-like development (Fig 5C). Interestingly, while the expression of TSLP remained increased in mutant[GFP] bulge HF-SCs and mutant[GFP] b-KCs at the mid-term of the psoriasis-like development, its

expression was reduced at the latest stage (D30) (Fig 5D and E). In contrast, an increased expression of TSLP in non-mutant[Tom] bulge HF-SCs was observed at D30, while non-mutant[Tom] b-KCs did not express TSLP at any stage of psoriasis-like development (Fig 5D and E). Furthermore, TSLP was increased in sera of both psoriasis-like mouse models, DKO* and DKO*[K15], at day 30 after psoriasis-like induction (Fig 5F). These results suggest that the secretion of TSLP may contribute to psoriasis-like initiation and maintenance in DKO* and DKO*[K15] mice. In summary, the RNA-seq analyses revealed the heterogeneity and complexity of different epidermal stem cell populations during psoriasis-like disease initiation and progression.

## TSLP secreted by mutant[GFP] keratinocytes induces hyper-proliferation of non-mutant[Tom] keratinocytes leading to epidermal hyperplasia

To mechanistically explore the effect of the pro-inflammatory TSLP protein on non-mutant[Tom] KC proliferation, we first induced the deletion of c-Jun and JunB in primary DKO*-mT/mG KC cultures by infecting cells with Adenovirus-Cre or empty Adenovirus (control). We observed that 24 h after infection, 50% of KCs were GFP[+] and deficient for c-Jun and JunB expression, while the other 50% were Tomato[+] KCs and maintained the expression of c-Jun and JunB. To assess the causal involvement of TSLP on non-mutant KCs proliferation, cells were treated with a blocking TSLP antibody for three consecutive days after Adenovirus infection (Fig 6A). The Adeno-Cre-infected cells significantly increased the amount of TSLP in conditioned media, when compared to Adeno-empty-infected control cells (Fig 6B). This correlated with a significant increase in the proliferation of non-mutant[Tom] KCs present in the Adeno-Cre-infected cell cultures, when compared to Adeno-empty control cells, while the hyper-proliferation was blocked by the addition of 1 μg/ml TSLP antibody (Fig 6C). In order to test whether TSLP is sufficient to induce proliferation in KCs, recombinant TSLP was added at different concentrations (Fig EV5A and B) to primary wild-type (WT) KCs. Primary WT KCs expressed TSLPR in culture and increased proliferation rate after recombinant TSLP treatment, even at low concentrations. Interestingly, primary WT KCs derived from bulge HF-SCs (CD34[+]) significantly increased the proliferation rate rather than basal KCs (CD49f[+]) in comparison with their control cells (Fig EV5C) suggesting that bulge HF-SCs are more sensitive to TSLP. These data show that the proliferation of non-mutant KCs is dependent on TSLP secretion from mutant KCs. TSLP exerts its biological effects by binding to a high-affinity heteromeric complex composed of TSLP receptor chain and IL-7Rα. Blocking TSLP led to down-regulation of TSLPR chain in co-cultures, while the IL-7rα chain increased possibly as a compensatory mechanism (Fig 6D and E). We also observed that VEGFα expression, a pro-inflammatory mediator in psoriasis, decreased with anti-TSLP treatment (Fig 6F). To determine the TSLP autocrine/paracrine regulation in co-cultures of mutant[GFP] with non-mutant[Tom] KCs, we purified by FACS GFP[+] and Tomato[+] KCs after TSLP neutralization and analyzed gene expression of pro-inflammatory mediators in psoriasis-like disease (Fig EV5D–M). We confirmed the reduction of TSLP receptor expression and overexpression of Il-7rα in both populations, mutant[GFP] and non-mutant[Tom] KCs, but statistically significant in mutant[GFP] KCs (Fig EV5E and F). In addition, VEGFα was increased in mutant[GFP] and non-mutant[Tom] KCs

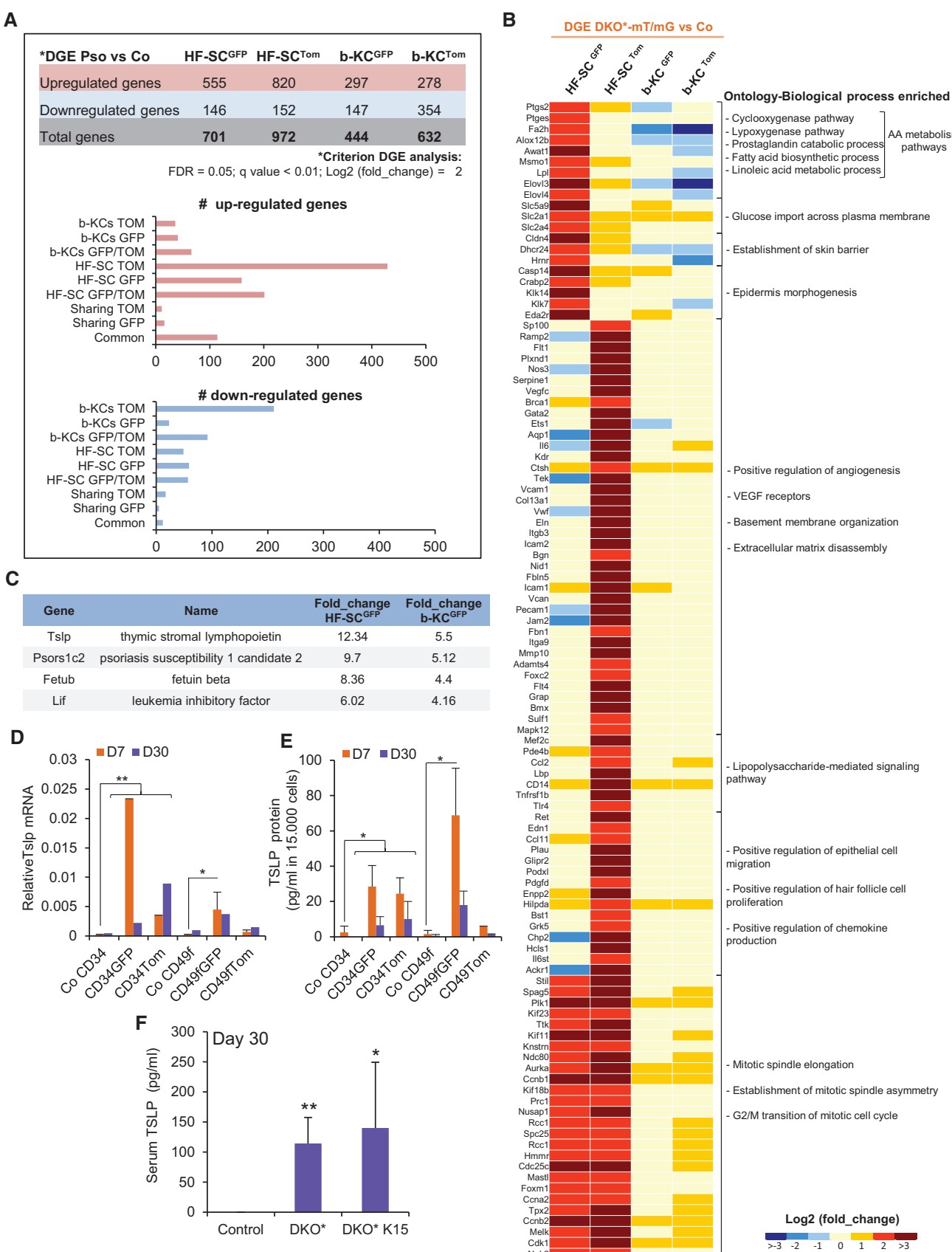

**Figure 5.**

◄

**Figure 5.  Different molecular signatures of mutant and non-mutant epidermal stem cells (EpSCs).**

A  Differentially expressed genes (DEGs) from the four subpopulations (bulge HF-SC[GFP]; bulge HF-SC[Tom]; b-KC[GFP]; and b-KC[Tom]) from psoriatic DKO*-mT/mG mice in comparison with their control counterparts (Co). The graphs represent the number of genes differentially up-regulated or down-regulated specifically in each group of interest obtained by Venn diagram analysis (Fig EV4B and C).

B  Heat map for specific genes relative to biological process enriched based on Gene Ontology IDs for DEGs that are: (i) exclusively up-regulated in HF-SC[GFP]; (ii) exclusively up-regulated in HF-SC[Tom]; (iii) commonly up-regulated in HF-SC[GFP] and HF-SC[Tom].

C  Prediction of secreted proteins enriched in GFP[+] subpopulations by the analysis of DEG commonly up-regulated in HF-SC[GFP] and b-KC[GFP] by ProteINSIDE analysis of RNA-seq.

D, E  Gene and protein expression of TSLP in different epidermal cell subpopulations sorted at mid-term (D7) or late term (D30) of psoriasis-like progression in DKO* mice. $n$ = 2–3. Data represent mean ± SD. Statistical significance *$P < 0.05$, **$P < 0.01$ (Student's two-tailed $t$-test relative to control groups). See Appendix Table S2 for exact $P$-values.

F  Quantification of TSLP production in sera of control, DKO*, and DKO*[K15] mice at day 30 of psoriasis-like disease by ELISA. $n$ = 4 per group. Data represent mean ± SD. Statistical significance *$P < 0.05$, **$P < 0.01$ (Student's two-tailed $t$-test relative to controls). See Appendix Table S2 for exact $P$-values.

Source data are available online for this figure.

after Adeno-Cre infection, while its expression was down-regulated after TSLP neutralization in both populations (Fig EV5G). Interestingly, Il-6 was increased only in non-mutant[Tom] KCs after the induction with Adeno-Cre and was reduced after blocking with TSLP antibody (Fig EV5H). Other pro-inflammatory cytokines and mediators, such as IL-1β and p65 (NF-κB), increased only in mutant[GFP] KCs after c-Jun/JunB deletion with the consecutive reduction after TSLP neutralization (Fig EV5I and J). Furthermore, IFN-γ or G-CSF was up-regulated in both populations, GFP and Tom KCs; however, only mutant[GFP] KCs responded to TSLP neutralization (Fig EV5K and L). S100A9 increased in both populations, higher in mutant[GFP] KCs, but its expression was not reduced after blocking TSLP (Fig EV5M). These data suggest that TSLP acts as autocrine and paracrine factor in the cross-talk between mutant and non-mutant KCs and regulates the expression of different pro-inflammatory mediators.

To determine whether local neutralization of TSLP reduces epidermal cell proliferation and psoriasis-like progression *in vivo*, intradermal injections of TSLP antibody were applied to the right ear of both mouse models, DKO* and DKO*[K15], 5 days after psoriasis-like induction (Fig 6G). Anti-TSLP treatment ameliorated psoriasis-like plaque formation, ear thickness, and trans-epidermal water loss (TEWL) in both mouse models, significantly in DKO* mice (Fig 6H–J) but not in DKO*[K15] mice (Appendix Fig S6A and B). In order to determine whether this disease amelioration was caused by the reduction of epidermal cell proliferation and/or by reduction in the inflammatory response, FACS analyses were performed to quantify proliferative epidermal cells (Fig 6K) and immune cells infiltration (Appendix Fig S6C and D) from the ear of DKO* and DKO*[K15] mice. Proliferating epidermal cells were significantly reduced in both mouse models after anti-TSLP treatment (Fig 6K), confirmed by a reduction of epidermal hyperplasia and proliferation by Ki67 expression in ear sections (Fig 6L). Furthermore, the number of neutrophils on the treated ears was reduced in DKO* mice, but less in DKO*[K15] mice (Appendix Fig S6C and D).

The JAK/STAT signaling pathway through TSLP/TSLPR has been widely studied for the activation of dendritic cells in allergies/inflammatory diseases such as atopic dermatitis or airway allergies (Rochman *et al*, 2010). We next analyzed the expression of STAT3 and STAT5 phosphorylation (p-STAT3, p-STAT5) by immunofluorescence in ear sections of psoriasis-like mice treated with anti-TSLP or IgG (Appendix Fig S6E). DKO* mice treated with anti-TSLP had decreased epidermal thickness and STAT5 phosphorylation, while p-STAT3 remained unaffected in both models, suggesting that TSLP acts through p-STAT5.

Overall, these data show that TSLP secretion by bulge HF-SCs and mutant basal keratinocytes stimulates neighboring epidermal cells to hyper-proliferate, activate p-STAT5 and induce expression of VEGFα and IL-6, contributing to the initiation of epidermal hyperplasia and the maintenance of chronic skin inflammation observed in psoriasis lesions. Autocrine regulation by TSLP secretion may activate pro-inflammatory mediators, such as NF-κB, IL-1β, and G-SCF in mutant bulge HF-SCs and mutant basal keratinocytes, pushing the inflammatory signaling toward psoriasis progression.

### TSLP increases in hair follicles and IFE of scalp psoriasis patients

Finally, we analyzed the expression levels of TSLP in human scalp psoriasis patients (Fig 7). TSLP was expressed along the ORS in homeostasis of non-lesional or healthy scalp, while IFE did not show any expression. In contrast, TSLP expression level was increased in lesional areas with psoriasis in both compartments, IFE and hair follicles (Fig 7A–C). Interestingly, bulge HF-SCs labeled with CD200 did not express TSLP in psoriasis patients, neither lesional nor non-lesional scalp, while sub-bulge ORS increased its expression in lesional HFs (Fig 7D and E).

## Discussion

It is well established that Jun/AP-1 transcription factors are involved in chronic skin inflammation. Reduced expression of JUNB/c-JUN in lesional regions of the inter-follicular epidermis (IFE) of psoriatic patients might be critical for the development of the disease (Zenz *et al*, 2005) despite the high variability in the expression of these factors (Haider *et al*, 2006; Park *et al*, 2009; Guinea-Viniegra *et al*, 2014). Using an epidermal lineage tracing approach in skin inflammatory mouse models with inducible epidermal deletion of c-Jun and JunB, we show that mutant bulge hair follicle stem cells (HF-SCs) are sufficient to initiate and maintain disease development, whereas mutant IFE cells, also initiators of the disease, are lost during psoriasis-like progression. The lesional psoriasis-like areas of the mouse IFE displayed a mosaic pattern of cells expressing c-Jun/JunB, which is highly reminiscent to the observed variability in expression of JUNB/c-JUN in psoriatic plaques of patients (Haider *et al*, 2006; Park *et al*, 2009; Guinea-Viniegra *et al*, 2014). Interestingly, the analyses of human scalp psoriatic patients show for the first time a significant reduction in the levels of c-JUN and JUNB specifically in bulge HF-SCs,

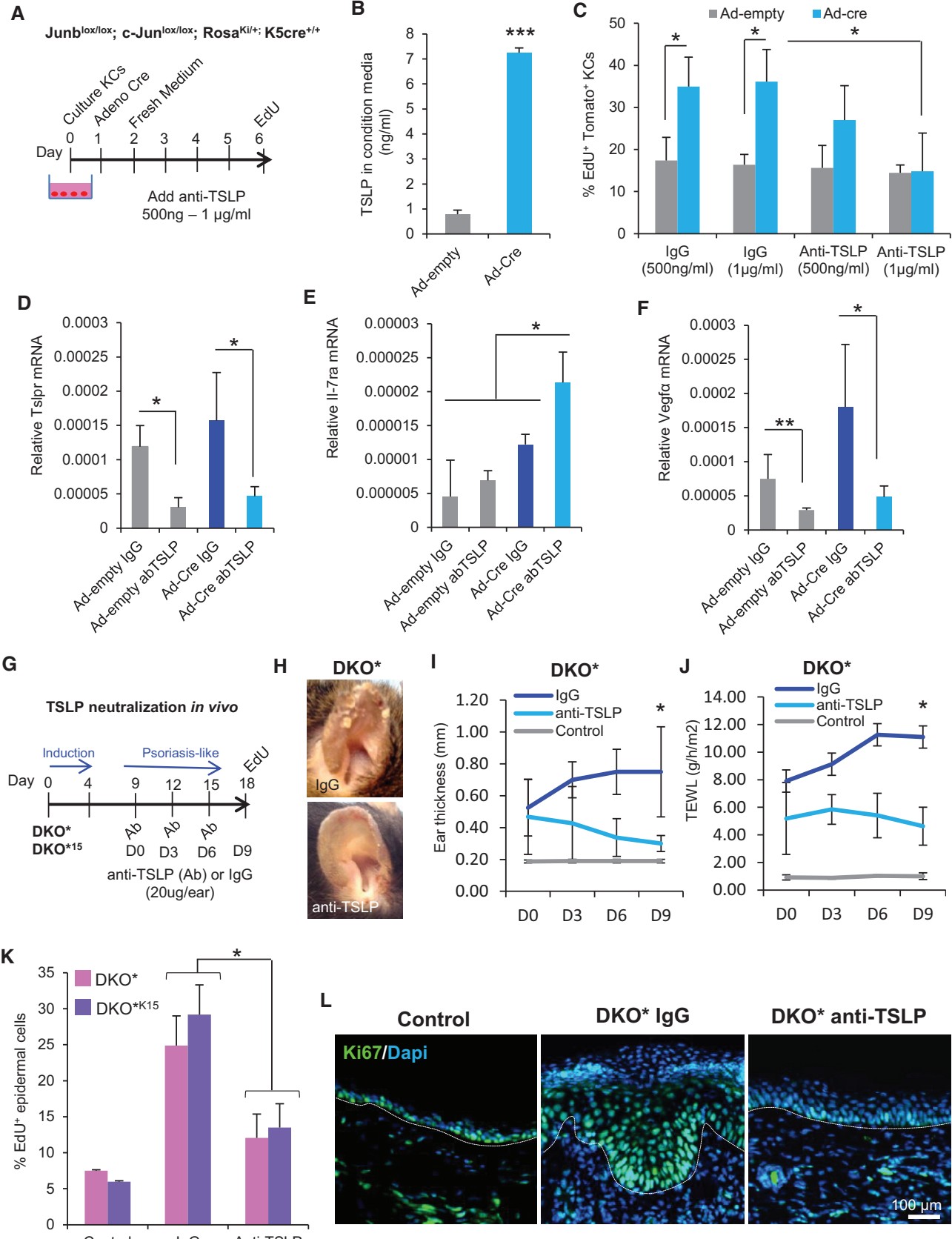

**Figure 6.**

Figure 6. TSLP secreted by mutant<sup>GFP</sup> epidermal cells induces hyper-proliferation of non-mutant<sup>Tom</sup> keratinocytes.

A Experimental design for induction of mutant$^{GFP}$ KCs *in vitro* and neutralization of secreted TSLP by anti-TSLP. Primary keratinocytes were cultured from the skin of Junb$^{lox/lox}$; c-Jun$^{lox/lox}$; Rosa$^{Ki/+}$; and K5cre-ERT$^{+/+}$ mice. 50% of mutant$^{GFP}$ KCs were induced by infection with Cre Adenovirus, the non-infected cells were non-mutant$^{Tom}$ KCs. Adeno-empty was used as control. EdU was added for 5 h to label proliferating Tomato$^+$ cells in 6-day co-culture. Quantification was performed by confocal imaging analysis.

B TSLP production in conditioned media quantified by ELISA three days after infection with Ad-empty or Ad-cre. $n = 3$ independent experiments. Data represent mean ± SD. Statistical significance ***$P < 0.001$ (Student's two-tailed $t$-test relative to control group). See Appendix Table S2 for exact $P$-values.

C Percentage of EdU$^+$ Tomato$^+$ keratinocytes after blocking with anti-TSLP. Anti-TSLP neutralizes the hyper-proliferation of non-mutant$^{Tom}$ KCs *in vitro* after infection with Ad-cre. $n = 3$ independent experiments, six fields per experiment. Data represent mean ± SD. Statistical significance *$P < 0.05$ (two-way ANOVA and Bonferroni post-test).

D Gene expression of Tslpr after TSLP neutralization is reduced. $n = 3$ independent experiments. Data represent mean ± SD. Statistical significance *$P < 0.05$ (Student's two-tailed $t$-test relative to control group). See Appendix Table S2 for exact $P$-value.

E Gene expression of IL-7ra after TSLP neutralization is increased. $n = 3$ independent experiments. Data represent mean ± SD. Statistical significance *$P < 0.05$ (Student's two-tailed $t$-test relative to control group). See Appendix Table S2 for exact $P$-value.

F Gene expression of VEGFα after TSLP neutralization is reduced. $n = 3$ independent experiments. Data represent mean ± SD. Statistical significance *$P < 0.05$, **$P < 0.01$ (Student's two-tailed $t$-test relative to control groups). See Appendix Table S2 for exact $P$-value.

G Experimental design for TSLP neutralization *in vivo*. TSLP neutralization was performed by triple intradermal injections of 20 μg of anti-TSLP or IgG to the right ears of DKO* and DKO*$^{K15}$ mice 5 days after psoriasis-like induction. EdU was added into mice by IP injection 2 h before euthanasia.

H Representative images of treated ears after 9 days of anti-TSLP or IgG treatment.

I, J Longitudinal analysis of ear thickness (I) and trans-epidermal water loss (TEWL) (J) in DKO* mice at different time points during psoriasis-like disease progression. Data represent mean ± SD. $n = 4$. Statistical significance *$P < 0.05$ (two-way ANOVA and Bonferroni post-test).

K Percentage of EdU$^+$ epidermal cells (CD45$^-$CD49f$^+$) after blocking with anti-TSLP in DKO* and DKO*$^{K15}$ mice. DKO* $n = 4$ and DKO*$^{K15}$ $n = 3$ per group. Data represent mean ± SD. Statistical significance *$P < 0.05$ (two-way ANOVA and Bonferroni post-test).

L Representative immunofluorescence images of ear sections from control or DKO* mice treated with IgG or anti-TSLP and stained for Ki67. Ki67 expression was reduced in mice treated with anti-TSLP ($n = 3$).

Source data are available online for this figure.

supporting our hypothesis that AP-1 transcription factors in the bulge may control HF-SC homeostasis.

The data from the mouse models suggest that the inducible deletion of c-Jun and JunB in HF-SCs can initiate and maintain psoriasis-like plaques in ear skin, the epidermal hyperplasia and inflammatory features of the disease, but it does not fully recapitulate the systemic effects observed in the DKO* psoriasis-like mice. These differences provide a unique opportunity to study separately the specific systemic features derived from mild/moderate psoriasis in DKO*$^{K15}$ mice vs. severe psoriasis-like disease in DKO* mice, including different target treatments in two complementary mouse models. Furthermore, these genetic modifications in IFE or HF-SCs might also reflect the high variability of the disease severity and psoriasis types in human patients, in which some patients show small psoriatic plaques with mild severity, while others develop multiple psoriatic plaques and co-morbidities.

An important finding using lineage tracing and cell dynamic studies in these mouse models is the distinct behavior of different epidermal stem cell populations during disease initiation and progression. Interestingly, the inflammatory skin disease is sustained mainly by non-mutant keratinocytes, while IFE mutant keratinocytes, basal and suprabasal, may contribute to the initiation, but not to the progression of the disease. Recent studies demonstrated that the epidermis is able to correct aberrant growth and inflammation to maintain tissue homeostasis through the elimination of damaged/mutant epidermal cells (Brown *et al*, 2017; Kashiwagi *et al*, 2017). However, in both psoriasis-like mouse models, although mutant IFE cells are eliminated over time, the chronic skin inflammation is perpetuated by the survival of mutant HF-SCs. We propose that the expression of c-Jun and JunB is differentially regulated in distinct stem cell populations in the epidermis and that specific disruptions in the levels of these AP-1 factors are important in HF-SCs, which enhance their survival and promote the development of psoriasis-like disease.

Importantly, mutant HF-SCs have an inflammatory transcriptomic signature distinct from other epidermal populations. For instance, several pro-inflammatory cytokines and arachidonic acid metabolic pathways described to be critical in psoriasis, such as IL-23 (Rizzo *et al*, 2011), IL-1α (Sanmiguel *et al*, 2009), TNF-α (Kock *et al*, 1990; Kumari *et al*, 2013), TSLP (Volpe *et al*, 2014), or PTGS2 (Samuelsson, 1991), were up-regulated. A recent report showed that during hair follicle regeneration, CD34$^+$ bulge HF-SCs displayed up-regulated expression levels of genes involved in the regulation of the cell cycle and inflammatory responses, such as TNF-α and IL-23 (Hoeck *et al*, 2017). The aberrant up-regulation of a pro-inflammatory signaling cascade in mutant HF-SCs may be sufficient to increase cell survival and the acquisition of a psoriasis-like phenotype, not only in mutant HF-SCs, but also in neighboring non-mutant cells. Interestingly, some arachidonic acid metabolites can regulate the survival of different adult stem cell populations (Widera *et al*, 2006; Porter *et al*, 2013). Furthermore, these pro-inflammatory mediators may also be involved in the recruitment of inflammatory cells, such as dendritic cells, neutrophils, or T cells (Albanesi *et al*, 2018).

Similarly, mutant epidermal cells can also stimulate neighboring non-mutant epidermal cells to trigger hyperplasia in a paracrine fashion. In both mouse models, we show that non-mutant keratinocytes acquire a proliferative advantage, which is induced by paracrine factors secreted by mutant keratinocytes such as TSLP. TSLP is a cytokine expressed by epithelial cells, including keratinocytes, and plays an important role in allergic inflammation, such as atopic dermatitis (AD) (He *et al*, 2008). Overproduction of TSLP was found in keratinocytes of human AD skin lesions (Soumelis *et al*, 2002) and in psoriasis patients (Volpe *et al*, 2014). Most studies focus on the role of TSLP in activating inflammatory cells, such as dendritic cells, although its role in epidermal cell–cell communication is not well understood. Our data indicate that TSLP secretion from mutant keratinocytes enhances cell proliferation and VEGFα expression in non-mutant keratinocytes, an effect that was found reduced by neutralizing TLSP. We have

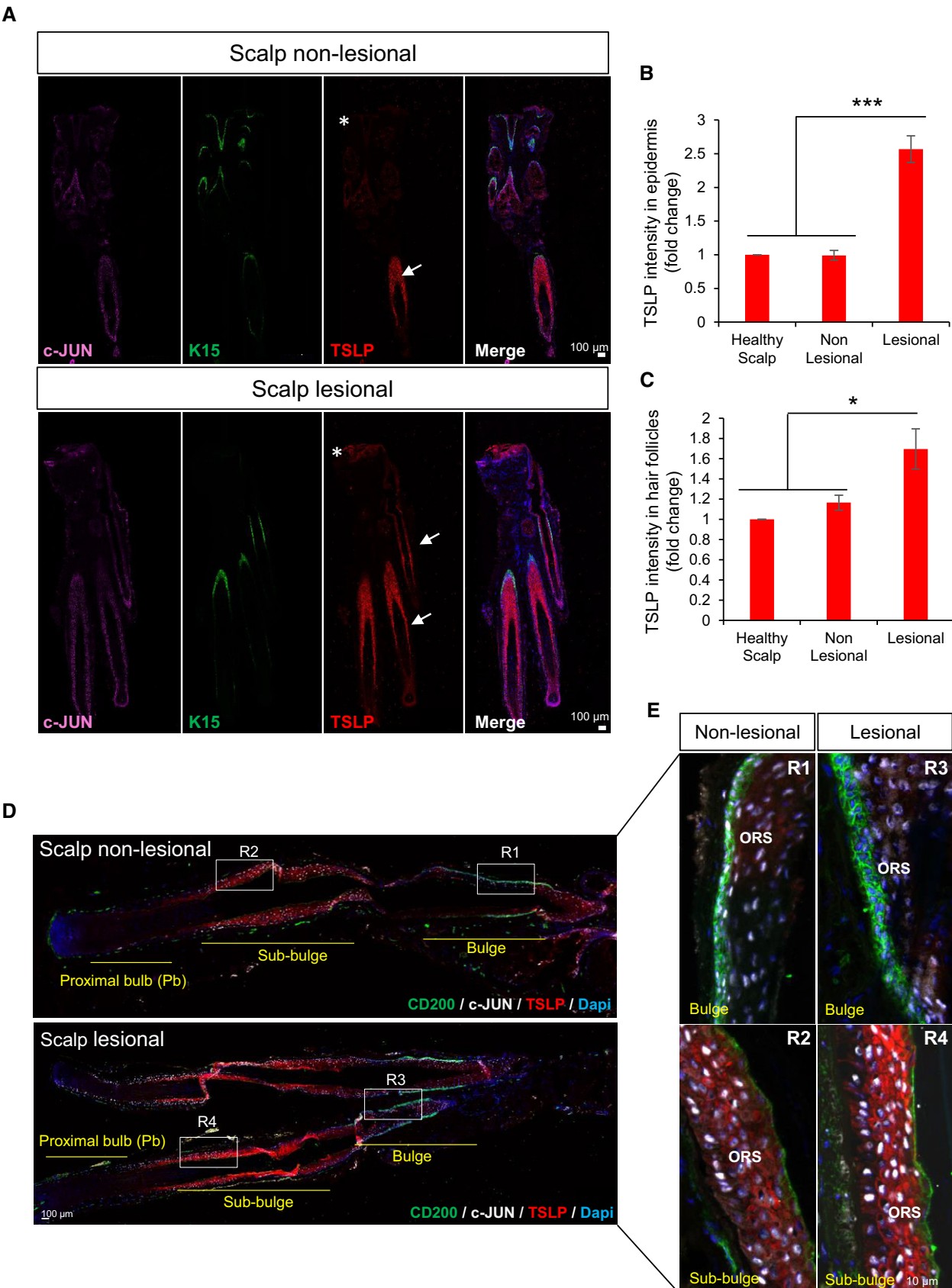

**Figure 7.**

**Figure 7.  Scalp psoriasis patient skin shows increased expression of TSLP in hair follicles and epidermis.**

A    Representative composite immunofluorescence images of whole hair follicle units by confocal from non-lesional and lesional scalp psoriasis patients. c-JUN (purple), K-15 (green), TSLP (red), and DAPI (blue). Asterisks represent the epidermis and arrows hair follicles.

B, C    TSLP intensity measurement in hair follicle units and epidermis by ImageJ from healthy scalp, non-lesional, and lesional scalp psoriasis patients. Representation of fold change relative to control healthy scalp. $n = 3$ samples. Data represent mean $\pm$ SD. Statistical significance *$P < 0.05$, ***$P < 0.001$ (Student's two-tailed $t$-test). See Appendix Table S2 for exact $P$-values.

D, E    Representative composite immunofluorescence images of whole hair follicle units by confocal from non-lesional and lesional scalp psoriasis patients (D) and magnification images of specific regions from the HFs (E). CD200 (green), c-JUN (white), TSLP (red), DAPI (blue).

previously reported in the DKO* mouse model that treatment with anti-VEGF antibody ameliorates the disease (Schonthaler *et al*, 2009) and epidermal-specific VEGFα overexpression in transgenic mice leads to spontaneous chronic skin inflammation (Xia *et al*, 2003). Importantly, during psoriasis-like disease initiation, TSLP is produced by mutant HF-SCs and at late stage also by non-mutant HF-SCs. These mutant HF-SCs also up-regulate several kallikreins (Klk7 and Klk14), and elevated Klk expression in keratinocytes results in significantly enhanced protease activity and augmentation of TSLP levels (Briot *et al*, 2010; Bin *et al*, 2011). Local administration of anti-TSLP in psoriasis-like mice resulted in the reduction of the phenotype due to epidermal hyperplasia regression and inhibition of STAT-5 phosphorylation (p-STAT5) in the epidermis. The activation of STAT-5 by TSLP has been described to play an important role in survival and proliferation of CD4$^+$ T cells in allergic/inflammatory skin and airway disorders (Rochman *et al*, 2010). However, phosphorylation of STAT3 in epidermal cells and immune cell infiltration remained equally activated in both mouse models, suggesting that epidermal proliferation induced by TSLP in psoriasis-like disease maybe acting through activation of STAT5, and not STAT3 phosphorylation. These novel findings extend the role of TSLP to the regulation of keratinocyte proliferation in psoriasis-like progression, which is an important hallmark in psoriasis. Interestingly, scalp psoriasis patients overexpressed TSLP in the IFE and in the ORS of the sub-bulge region of HFs; however, bulge HF-SCs did not express TSLP. CD34 labels sub-bulge cells in human HFs, but bulge HF-SCs are negative for this marker and positive for CD200 (Purba *et al*, 2014). Whether CD34 sub-bulge populations in human HFs or other stem cell populations respond to TSLP is still unsolved. The specific response to TSLP by different stem cell subpopulations in the hair follicles of psoriatic patients as well as potential cross-talk between cell populations will require further investigation. In addition, further investigations will be required to determine whether local neutralization of TSLP in combination with specific HF-SCs pro-inflammatory mediators such as PGE2 could be an effective treatment for psoriasis.

In conclusion, our study describes a novel contribution of epidermal stem cells in psoriasis-like development and provides insights into new cellular and molecular mechanisms underlying the disease. These findings open new avenues for further investigations toward discovering novel therapies against psoriatic HF-SCs, which might be beneficial for treating specifically psoriasis from different skin regions.

# Materials and Methods

## Human psoriasis biopsies

A total of seven scalp human biopsies were obtained in this study. Five were obtained from psoriatic patients with lesional scalp psoriasis and non-lesional scalp psoriasis from the same patient, and two healthy scalp samples from occipital scalp region of volunteer hair transplant patient. All samples were obtained through informed consent after protocol approval by the Clinical Research Ethical Committees of Hospital Donostia and the University of Las Palmas de Gran Canaria, and the experiments conformed to the principles set out in the WMA Declaration of Helsinki and the Department of Health and Human Services Belmont Report. Scalp biopsies were harvested using between 1 and 4 mm diameter punch (at least with a hair follicle unit). Scalp biopsies were directly embedded in optimal cutting temperature (OCT) compound for frozen sections, or fixed 24–48 h with neutral buffered 4% paraformaldehyde (PFA) and directly paraffin-embedded. Five-micrometer sections were stained either with hematoxylin and eosin (H&E) or processed for immunofluorescence staining (see below).

## Generation of lineage tracing system in psoriasis-like mouse models

All mouse experiments were performed in accordance with the royal decree D53/2013 on the protection of animals used for scientific purposes (DIRECTIVE 2010/63/EU), and the Ethical Committee of the Health Institute Carlos III, "authorized Body" by the *Comunidad de Madrid* (CAM), has evaluated and approved all the procedures involving animal experimentation for this project.

The generation of the DKO* psoriasis-like mouse model has previously been described (Zenz *et al*, 2005). Briefly, DKO* psoriasis-like mouse model with mix background was generated by the cross of mice carrying the floxed *JunB* allele (*JunB*$^{f/f}$) and floxed *c-Jun* allele (*c-Jun*$^{f/f}$) with the transgenic mice expressing the Cre recombinase–estrogen receptor fusion under the control of the basal keratinocyte-specific K5 promoter (K5-Cre-ERT) to obtain *JunB*$^{f/f}$ *c-Jun*$^{f/f}$ K5-Cre-ERT mice. DKO*$^{K15}$ psoriasis-like mouse model was generated by the cross of mice carrying the floxed *JunB* allele (*JunB*$^{f/f}$) and floxed *c-Jun* allele (*c-Jun*$^{f/f}$) with the transgenic mice expressing the Cre recombinase–prostaglandin receptor fusion under the control of the bulge hair follicle stem cell-specific keratin 15 promoter (K15-Cre-PGR) to obtain *JunB*$^{f/f}$ *c-Jun*$^{f/f}$ K15-Cre-PGR mice.

To trace epidermal-specific deletion of *JunB*$^{f/f}$ and *c-Jun*$^{f/f}$ in DKO* and DKO*$^{K15}$ mice, global double-fluorescent Cre reporter mouse Gt(ROSA)26Sor$^{tm4(ACTB-tdTomato,-EGFP)Luo/J}$ (referred as mT/mG: membrane-Tomato/membrane-Green) was crossed with DKO* and DKO*$^{K15}$ mice (referred as DKO*-mT/mG and DKO*$^{K15}$-mT/mG). Mice expressing Cre recombinase under the control of the K5 promoter or K15 promoter with wild-type (+/+) or heterozygous (+/f) for *JunB* and *c-Jun* were used as controls (Co$^{K5}$-mT/mG and Co$^{K15}$-mT/mG, respectively). Eight-week-old mutant mice and controls (males and females) were injected daily (intraperitoneal),

four times with 2 mg tamoxifen (Sigma) or mifepristone (Cayman Chemical) in DKO*-mT/mG and DKO*[K15]-mT/mG to induce the deletion of floxed *JunB* and *c-Jun* alleles in the DKO*-mT/mG and DKO*[K15]-mT/mG, respectively. In addition, constitutive Tomato expression was irreversibly replaced by the expression of GFP upon the removal of the loxP-flanked stop sequence only in the basal layer of epidermal cells that delete *JunB* and *c-Jun* in DKO*-mT/mG, or only in bulge hair follicle stem cells in the case of DKO*[K15]-mT/mG animals.

## Intravital imaging

Psoriasis-like DKO*-mT/mG mice at day 7 after tamoxifen induction were anaesthetized with vaporized isoflurane and placed in a hand-made platform for the mounting of the ear into a coverslip until it was completely flattened and immobile. Mice were well hydrated by a subcutaneous injection of 1 ml saline, and their temperature was kept at 35°C in a microscope incubator equipped with an air heater and temperature probe (The Cube, Life Imaging Services). Mice were provided with vaporized isoflurane at 1.5% through a nose cone for the course of the live imaging session during 8 h.

All mice were imaged with a TCS-SP5 confocal microscope (Leica Microsystems) equipped with AOBS and a HCX PLAN APO 20×/0.7 N.A. dry objective. EGFP and TdTomato emission were acquired every 10 min. Image processing, measurements, assembly, and editing of time-lapse movies were performed using Image J.

## Immunofluorescence

Serial sections from human scalp biopsies and mouse ears embedded in OCT or human scalp biopsies embedded in formalin-fixed paraffin were processed for immunofluorescence. Frozen sections were fixed with neutral buffered 4% paraformaldehyde (PFA) for 15 min at room temperature (RT) and washed three times with phosphate-buffered saline (PBS). Paraffin sections were processed for their deparaffinization and antigen retrieval. Primary antibodies specific for c-Jun (Cell Signaling, clone 60A8, ref # 9165, dilution 1:100), JunB (Cell Signaling, clone C37F9, ref # 3753, dilution 1:100), Keratin 15 (Santa Cruz, clone LHK15, dilution 1:400), CD200 (Bio-Rad, ref # MCA1960GA, dilution 1:100), GATA-6 (R&D Systems, ref # AF1700, dilution 1:400), TSLP (BioLegend, ref # 515901, dilution 1:200), TSLPR (Invitrogen, ref # Pa5-20380, dilution 1:200), Keratin 5 (BioLegend, ref # 905901, dilution 1:1,000), cleaved caspase-3 (Cell Signaling, clone Asp175, ref # 9661S, dilution 1:100), Ki67 (Master Diagnostica, clone SP6, dilution 1:5), pSTAT3 (Cell Signaling ref # 9145, dilution 1:100), and pSTAT5 (Cell Signaling ref # 9351, dilution 1:100) were incubated overnight (O/N) at 4°C. Sections were then washed three times with PBS and incubated with secondary antibodies conjugated to the appropriate fluorophores (Alexa Fluor 647, Life Technologies, dilution 1:200) for 1 h at RT. Sections were then washed three times with PBS and mounted with gel mount to be analyzed by fluorescence microscopy or confocal.

For whole-mount immunostaining, ear skin was dissected, fixed in 4% paraformaldehyde O/N at 4°C, and washed three times in PBS. The samples were incubated in 1.5% Triton X-100 and 0.1% Tween in PBS (PBS-TT) for 3 h at RT. Then, samples were blocked with 10% donkey serum in PBS-TT (blocking buffer) O/N at RT.

Primary antibody for staining bulge hair follicle stem cells, CD34 (BD Pharmingen™, clone RAM34, ref # 553731, dilution 1:100), was diluted in blocking buffer and incubated 24 h at RT. After the staining, samples were washed three times during 6 h in PBS-TT. For immunofluorescence detection, the samples were incubated in blocking buffer containing Alexa-647-conjugated secondary antibodies for 24 h at RT. After three washes as described previously, the samples were stained with Hoechst (1:500) O/N at 4°C. Three washes with PBS-TT were performed, and samples were kept in glycerol at 80° for confocal microscopy imaging.

## Flow cytometry analysis and FACS

To obtain fresh dissociated epidermal cells from ear skin of psoriasis-like and control mice, the epidermis was separated from the dermis by trypsin digestion (0.75%) for 1 h at 37°C. After mechanical dissociation, the cell suspension was filtered through a 70 μm cell strainer and centrifuged 5 min at 445 g. Pellet was re-suspended in culture medium for further flow cytometry analysis or FACS depends on experiment.

Analysis of different epidermal subpopulations by flow cytometry was carried out. Epidermal cells were counted, centrifuged and re-suspended in PBS with 1% chelated FBS and 1% BSA (staining buffer), and stained for 30–45 min at 4°C with the following conjugated surface markers depending on the experiment: CD45-APC-Cy7 (BioLegend, clone 30-F11, dilution 1:400), CD31-PerCP/Cy5.5 (BioLegend, dilution 1:100), CD49f-PE-Cy7 (BioLegend, dilution 1:400), CD34-Alexa-647 (BD Pharmingen™, clone RAM34, dilution 1:100), Sca-1-PerCP/Cy5.5 (BioLegend, clone D7, dilution 1:400), CD3e-PE-Cy7 (BioLegend, dilution 1:100), CD11b-FITC (BioLegend, dilution 1:100), CD11c-APC (BioLegend, dilution 1:100), and Ly6G-PeRP/Cy5.5 (BioLegend, dilution 1:100). EdU staining was carried out following the manufacturing protocol (Click-iT Plus EdU Alexa Fluor 647 Flow cytometry Assay Kit, Molecular probes, # C10634).

After staining, cells were washed with staining buffer, stained with DAPI to exclude dead cells, and analyzed by FACS CANTO II (BD, San Jose CA). At least 25,000 alive single events were collected. All data are analyzed using FlowJo 7.6.5 (Treestar, Oregon).

Sorting of different epidermal subpopulations (basal keratinocytes and hair follicle stem cells) was performed after carrying out the staining explained above. After staining, cells were filtered through a 40 μm cell strainer onto 5 ml polypropylene sterile tubes. Sorting procedures were done excluding dead cells and doublets. GFP+ and Tomato+ epidermal basal keratinocytes and hair follicle stem cells were purified into TRIzol LS for RNA isolation, or into cell lysis buffer 1× (Cell Signaling, ref # 9803S) for ELISAs, or collected in keratinocyte serum-free medium with growth factors (KGM, Invitrogen) for primary keratinocyte cultures.

## Primary cultures and live-cell imaging

For keratinocyte colony formation assay, basal keratinocytes and hair follicle stem cells sorted by FACS were cultured in a 12-well culture plate treated with Coating Matrix Kit (Cascade Biologicals) in KGM (Invitrogen) in a concentration of 500 cells/cm². After 10 days in cultures, Tomato+ and GFP+ colonies of keratinocytes were quantified by fluorescence microscopy. Isolated keratinocytes

and keratinocyte colonies formed by less of 10 cells were discarded in this study. Three independent experiments with technical duplicates were performed for statistical analysis.

For live-cell time-lapse imaging of purified single cultures or co-cultures of Tomato$^+$ and GFP$^+$ keratinocytes, fresh isolated epidermal cells from ear skin of psoriatic-like DKO* or control mice were cultured in KGM in a 24-well culture plate treated with Coating Matrix Kit in a concentration of 2,000 cells/cm$^2$. Each type of epidermal subpopulation was cultured alone, or in contact co-cultures (1:1 proportion) of mutant$^{GFP}$ cell with non-mutant$^{Tom}$ cells or in co-cultures using 12-well trans-well system, in which non-mutant$^{Tom}$ cells were cultured in the bottom and mutant$^{GFP}$ cells were cultured in the insert. Time-lapse images were captured using CCD confocal incubator microscope at 37°C and 5% CO2. Images of small colonies formed after 3–5 days in culture from alone GFP$^+$ epidermal cells, or co-cultures of GFP$^+$: Tomato$^+$ keratinocytes were captured with 20× objective as six selected fields where keratinocyte colonies were located in 15-min intervals for 48 h. Images of Tomato$^+$ keratinocytes from trans-well experiments were captured each 48 h during 6 days. Cell death was analyzed by TOPRO-3 and DAPI staining. Image processing, measurements, assembly, and editing of time-lapse movies were performed using Image J. Three independent experiments were performed for statistical analysis.

To recombinant TSLP treatment, fresh isolated epidermal cells from ear and tail skin or purified bulge HF-SCs and basal KCs of WT mice were cultured in KGM in a 24-well culture plate treated with Coating Matrix Kit in a concentration of 10,000 cells/cm$^2$. Next day, mouse recombinant TSLP (eBioscience, ref # 14-8498-80) was added at different concentrations during 48 h. 5-Ethynyl-2′-deoxyuridine (EdU) was added to the cultures after 2 days with mouse recombinant TSLP in a concentration of 10 µM during 5 h at 37°C. EdU and anti-TSLPR staining was performed, and random confocal images were captures. EdU$^+$ keratinocytes were quantified by Image J. Three independent experiments were performed for statistical analysis.

**Adenovirus infection and TSLP neutralization**

Fresh dissociated epidermal cells from ear and tail epidermis of adult *JunB$^{f/f}$ c-Jun$^{f/f}$ RosamT/mG$^{f/f}$* mice were plated in a 24-well culture plate treated with Coating Matrix Kit (Cascade Biologicals) in KGM (Invitrogen) in a concentration of 15,000 cells/cm$^2$. Twenty-four hours later, *JunB$^{f/f}$ c-Jun$^{f/f}$ RosamT/mG$^{f/f}$* keratinocyte cultures were infected with adenoviruses expressing Cre (Ad-cre) or empty (Ad-empty) purchased from the University of Iowa in a concentration of $3 × 10^6$ particles of virus per ml of KGM. The medium with adenoviruses was changed by fresh KGM 16 h after infection. Two days after infection, around 50–60% of primary keratinocytes became GFP positive and the rest were Tomato positive and anti-TSLP antibody (PROALT-Protein Alternatives S.L, clone 28F12) was added to the cells every day during three consecutive days in a concentration of 500 ng/ml or 1 µg/ml. Mouse IgG was added in parallel as control. 5-Ethynyl-2′-deoxyuridine (EdU) was added to the cultures after 3 days with anti-TSLP antibody in a concentration of 10 µM during 5 h at 37°C. For further analysis, cells were fixed with PFA 4% in PBS for EdU staining (Thermo Fisher, C10086) or they were harvested for RNA isolation from total

cultures or from purified GFP$^+$ and Tom$^+$ KCs by FACS. Conditioned medium was also harvested to quantify soluble TSLP by ELISA according to the manufacturer's instructions (Mouse TSLP Quantikine ELISA Kit, Ref # MTLP00, R&D Systems). Conditioned medium was diluted to 1:50.

*In vivo* neutralization of TSLP was performed by three intradermal injections of 20 µg of ab-TSLP or IgG each 72 h on the right ear of DKO* and DKO*[15] 5 days after psoriasis-like induction. Left ears were injected with PBS. EdU was applied to mice by IP injection 2 h before euthanasia. FACS analyses were performed as described above.

**RNA isolation and qPCRs**

After FACS, each purified epidermal subpopulation was collected in TRIzol LS (Thermo Fisher Scientific) for further RNA isolation as described by manufacturer's instructions. RNA extraction was followed by reverse transcription using Promega kit (A2801). qPCR analyses were carried out using SYBR Green Master Mix Kit (Qiagen, 204143). Expression levels were compared to housekeeping gene RPL4. The primer sequences are shown in Appendix Table S1.

**RNA-seq**

Whole-genome transcriptomic analysis was performed by Next-Gen RNA-sequencing in four groups of psoriatic-like epidermal subpopulations after FACS and RNA isolation described above: Mutant$^{GFP}$, non-mutant$^{Tom}$ basal keratinocytes, mutant$^{GFP}$, and non-mutant$^{Tom}$ hair follicle stem cells were purified from the ears of three independent psoriatic-like DKO* mice after 7 days of tamoxifen induction. RNA-seq was also performed in control basal keratinocytes and hair follicle stem cells from the ears of three Co-mT/mG mice for further analysis. Next-Gen library construction and sequencing procedures were carried out by the Genomic Unit at CNIO, and further analysis was carried out by Bioinformatics Unit at CNIO. After confirmation of RNA quality by Agilent 2100 Bioanalyzer, variable amounts of total RNA samples, between 4 and 10 ng, were processed with the SMART-Seq v4 Ultra Low input RNA Kit (Clontech) by following manufacturer's instructions. Briefly, oligo(dT) primes preferentially reverse transcription of poly(A) RNA in the presence of a template switching oligonucleotide, and then, limited-cycle PCR produces double-stranded cDNA. Resulting cDNA was subsequently processed with the "Nextera XT DNA Library Prep Kit" (Illumina), which fragments and inserts Illumina adapters. Adapter-ligated non-directional libraries were completed by 12 cycles of PCR.

The resulting purified cDNA library was applied to an Illumina flow cell for cluster generation and sequenced on an Illumina HiSeq 2500 following manufacturer's protocols. Fifty-one base single-end sequencing reads were analyzed with the nextpresso pipeline (Grana *et al*, 2017), as follows: Sequencing quality was checked with FastQC v0.11.0 (http://www.bioinformatics.babraham.ac.uk/projects/fastqc/). Reads were aligned to the mouse genome (NCBI37/mm9) with TopHat-2.0.10 (Trapnell *et al*, 2012) using Bowtie 1.0.0 (Langmead *et al*, 2009) and SAMtools 0.1.19 (Li *et al*, 2009), allowing two mismatches and 20 multihits. Differential expression was tested with Cufflinks (Trapnell *et al*, 2012), using the mouse NCBI37/mm9 transcript annotations from https://ccb.jhu.edu/software/tophat/igenomes.shtml. GSEAPreranked (Subramanian *et al*, 2005) was used to perform gene set enrichment analysis of the

described gene signatures on a pre-ranked gene list, setting 1,000 gene set permutations. Differentially expressed genes (DEGs) were further analyzed for Venn's diagram (Venny[2.1], http://bioinfogp. cnb.csic.es/tools/venny/), Gene Ontology Biological Process (Enrichr, http://amp.pharm.mssm.edu/Enrichr/), and predictable secreted proteins (ProteINSIDE, http://www.proteinside.org/).

### Statistical analyses

For studies including animals, $n$ is the number of total animals tested, and for *in vitro* studies, the number of independent experimental replicates is indicated in Figure legends, with $n$ representing number of repeats with different sets of cells. For RNA-seq, three control mice and 3 DKO* mice were used. Data in bar graphs represent mean ± standard deviation (SD). Statistical analyses were performed using Prism5 (GraphPad) software. When comparing two groups, a two-tailed Student's *t*-test was used, and to compare three or more groups, a two-way ANOVA and Bonferroni post-test was implemented. *P*-values were calculated with a confidence interval of 95 % to indicate the statistical significance between groups. Statistically significant differences between groups are noted in figures with asterisks (*$P < 0.05$, **$P < 0.01$, ***$P < 0.001$).

## Data availability

All primary data, detailed protocols, and non-commercially available materials can be requested from the corresponding author. The RNA-seq datasets have been deposited to GEO under the accession number GSE119762 (https://www.ncbi.nlm.nih.gov/geo/query/acc.cgi?acc=GSE119762).

**Expanded View** for this article is available online.

## Acknowledgements

We thank Drs. M. Serrano and M. Pérez-Moreno for the Gt(ROSA)26Sor[tm4(ACTB-tdTomato,-EGFP)Luo/J] and K15-Cre-PGR mouse lines. We are very grateful to Drs. M. Pérez-Moreno, F. Real, Ö. Uluçkan, L. Bakiri and the laboratory members of the Sibilia and Wagner groups for critical reading of the manuscript and valuable suggestions. We thank V. Bermeo, G. Medrano, S. Leceta, O. Graña, and M. Pérez for their technical help and IT support. We acknowledge R. Paus laboratory members for the shipment of hair follicle samples. N.G.L. received funding from the People programme (Marie Curie Actions) of the European Union's Seventh Framework Programme (FP7/2007-2013) under REA grant agreement no 608765. A.I is funded by the Institute of Health Carlos III (PI16/01430). The Wagner laboratory was funded by a grant from the Spanish Ministry of Economy and competitiveness (SAF2015-70857RE, cofounded by the European Regional Development Fund) and is supported by the ERC (ERC-AdG 2016 CSI-Fun).

## Author contributions

NG-L conceived the project, designed and performed the experiments, and wrote the article. LFM performed experiments and contributed to manuscript writing. DM analyzed confocal microscopy experiments. GM-S analyzed RNA-seq. AI and FJ provided the human samples. EFW supervised the project, provided funding, and edited the article.

## Conflict of interest

The authors declare that they have no conflict of interest.

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
