## [Review Process File · EMBO Molecular Medicine]

Role of bulge epidermal stem cells and TSLP signaling in psoriasis

Nuria Gago-Lopez, Liliana F. Mellor, Diego Megías, Guillermo Martín-Serrano, Ander Izeta, Francisco Jimenez and Erwin F. Wagner

Review timeline:

Submission date:	1 April 2019
Editorial Decision:	29 April 2019
Revision received:	11 July 2019
Editorial Decision:	9 August 2019
Revision received:	27 August 2019
Accepted:	29 August 2019

Editor: Lise Roth

Transaction Report:

1st Editorial Decision

29 April 2019

Thank you for the submission of your manuscript to EMBO Molecular Medicine. We have now heard back from the 3 referees whom we asked to evaluate your manuscript.

As you will see from the reports below, they acknowledge the potential interest and translational relevance of the findings, however they also have fundamental concerns that should be addressed in a major round of revision of the present manuscript.

Addressing the reviewers' concerns in full will be necessary for further considering the manuscript in our journal. EMBO Molecular Medicine encourages a single round of revision only and therefore, acceptance or rejection of the manuscript will depend on the completeness of your responses included in the next, final version of the manuscript.

Please also contact us as soon as possible if similar work is published elsewhere. If other work is published, we may not be able to extend the revision period beyond three months.

I look forward to receiving your revised manuscript.

***** Reviewer's comments *****

Referee #1 (Comments on Novelty/Model System for Author):

I believe the lineage tracing studies are well done and the in vitro studies indeed reveal intriguing means by which mutant and non mutant stem cells interact and drive psoriasis. This study also raises

the possibility of targeting TSLP in psoriasis or stem cell derived factors in disease.

Referee #1 (Remarks for Author):

In their Article "Role of epidermal stem cell populations and TSLP signaling in inflammatory skin disease", Gago Lopez and colleagues examine the contribution of distinct epithelial stem and progenitor subsets to psoriasis like skin inflammation. The use tissue specific loss of cJun/JUNB to assess the contribution distinct progenitors to disease induction and inflammatory pathology. In doing so they uncovered that mutant cells lacking cJun/JUNB can influence the behavior of neighboring non-mutant cells, building upon their previously published idea that loss of cJun/JUN in epithelia is sufficient to induce inflammation. They Identify TSLP as a secreted mediator responsible for communication between mutant and non-mutant cells

This is an interesting article and well suited for EMBO Molecular Medicine. I have the following concerns that should be addressed prior to publication.

- 1) For lineage tracing students in figure 2 and 3 in many panels the D0 timepoints are missing, it's important to have these data as they indicate the initial cells that are markers and may change between D0 and D5 (presented)
- 2) Figure 4B the authors make the conclusion that HFSCs are less prone to cell death, however information about confluence of cultures is not included or discussed. Are basal KCs which clearly proliferate at a higher rate reaching confluences sooner and thus also more prone to cells death. Along those lines it is not clear in 4F why co-culture pf GFP+ cells with DKO cells results in lower numbers than in 4E. Also does panel G use HF-SCs or KCs for their studies. These issues need to be clarified/addressed in the legends.
- 3) It is unclear why the authors choose to show Ki67 in 6L and quantify Edu cells in 6K. Since Edu is a more faithful marker of proliferating cells, this should be used but for image analysis and quantification.
- 4) Is TSLP sufficient to induce proliferation in tomato+ KCs? KCs from WT mice? In other words, are other factors from the GFP+ cell environment priming the non-mutant cells? Along those lines the authors assay for TSLPra but not IL7ra which is also required for TSLP signaling.
- 5) In general, the legends could use more clarity and details. The figures are often hard to understand based on the results and legends.

Referee #2 (Comments on Novelty/Model System for Author):

1. The methodology is appropriate and the data shown convincing.
2. The authors indicate that the decreased expression of c-Jun and JunB in hair follicle stem cells is sufficient for psoriasis-like disease. This is quite novel.
3. Psoriasis is a common inflammatory disease in the skin. Understanding the mechanisms of psoriasis development will help the treatment of patients.
4. The authors used model mice adequately.

Referee #2 (Remarks for Author):

Psoriasis is a common inflammatory disease in the skin. Understanding the mechanisms of psoriasis development will help the treatment of patients.

In this article, Gago-Lopez et al found the decreased expression of c-JUN and JUNB in the bulge region of hair follicles, known as a hair follicle stem cell niche, in psoriatic patients. The authors then confirmed that decreased expression of c-Jun/JunB in hair follicle stem cells is sufficient for the initiation of psoriasis-like disease using mutant mice previously generated by the authors. The authors also identified TSLP as a paracrine factor that is secreted from mutant hair follicle stem cells and mutant basal keratinocytes. TSLP induces proliferation of neighboring non-mutant keratinocytes and VEGF expression, which results in the development and maintenance of a psoriasis-like

inflammatory skin disease.

This is an interesting study with clear medical relevance. The methodology is appropriate and the data shown convincing. Altogether, I recommend publication of this manuscript in EMBO Molecular Medicine pending the authors address the following comments.

Major comments

This manuscript starts with the human clinical samples where c-JUN and JUNB are decreased in the bulge region of hair follicles in psoriatic patients (Fig. 1). And the latter of this study, the authors identify TSLP as a mediator of psoriasis-like disease in c-Jun/JunB mutant mice. As the mouse model study indicated, is TSLP expression increased in the bulge region of hair follicles in psoriatic patients? The authors should confirm the expression of TSLP in psoriatic patients and indicate the results in the manuscript.

In Fig. 2I, the authors describe that mutant GFP⁺ keratinocytes are maintained around hair follicles. But it is very hard to confirm the location of GFP-positive cells. The authors should also indicate the histological sections.

In Fig. 4B, the authors indicate that HF-SC-derived mutantGFP KCs show long-term proliferative capacity and mutantGFP b-KCs show only short-term proliferative capacity. But, the stem cell population is enriched in the culture of HF-SC population-derived KCs. In contrast, the culture of b-KCs consists of keratinocytes with a variety of growth potentials. This experiment simply shows the difference of proliferative capacity between stem cell and transient amplifying cell populations. Therefore, the authors cannot claim that HF-SC-derived mutant KCs has different properties in proliferation and survival using this experimental model.

In Fig. 5C, the authors indicate the 4 genes coding for secreted proteins and then investigated TSLP in detail. Are other 3 genes not involved in the development of psoriasis? Otherwise, the authors should simply mention and/or discuss about that.

In Fig. 6H, the authors clearly demonstrate that anti-TSLP inhibits psoriasis-like disease development. Does TSLP neutralization also alleviate psoriasis-like disease after the development of this disease? The authors indicate that mutant b-KCs are eliminated in the late -stage but mutant HF-SCs are maintained and contribute to the chronic skin inflammation. But the authors did not show any evidence that mutant HF-SC contributes to the maintenance of this disease.

Minor comments

In page 8, line 1-3, the authors describe the induction of GFP with tamoxifen treatment. This result is indicated in EV 1A, right panels, but not EV 1B, upper panels.

In page 17, lines 2-3, the authors describe that "displayed a 12-fold increase in mutantGFP HF-SCs and a 4-fold increase ...". But in Fig. 5C, the fold change in HS-SCGFP is 6.17 and that of b-KCGFP is 2.75. how do readers interpret this?

Referee #3 (Remarks for Author):

In the manuscript 'Role of epidermal stem cell populations an TSLP signaling in inflammatory skin disease' by Gago-Lopez et al., the authors claim that hair follicle stem cells are sufficient to initiate psoriasis-like disease in murine skin. Their major findings are: 1) levels of c-Jun and JUNB are reduced in hair follicle stem cells from human psoriatic patients; 2) Lineage tracing studies revealed that IFE keratinocytes that are depleted of c-Jun and JUNB exhibit different proliferation and apoptosis rates in psoriasis-like disease; 3) Deletion of c-Jun and JUNB specifically in the K15⁺ population of HFSCs results in the development of a psoriasis-like phenotype that is milder as the inflammatory phenotype observed in mice lacking c-Jun and JUNB in all keratinocytes; 4) secreted factors from keratinocytes that lack c-Jun and JUNB induce hyperproliferation in wild-type neighboring epidermal cells and molecular signatures differ between both cell types; 5) secretion of TSLP

by keratinocytes that lack c-Jun and JUNB induces hyperproliferation of keratinocytes and blocking TSLP in vivo and in vitro ameliorates the inflammatory features that are linked to deletion of c-Jun and JUNB from HFSCs.

This is an interesting study showing a role for c-Jun and JUNB specifically in HFSCs, however there are various concerns as several claims are not properly backed up by experimental evidence.

Major concerns:

1. All in vivo studies have been conducted on ear skin. Hair follicles in hyperplastic ear skin are often not clearly distinguishable from the IFE (see Figure 2C, H and I; Figure 3). These analyses should be repeated on tail skin, where the different stem cell populations are spatially restricted and the difference between IFE and HFSCs is much more obvious. Many of the major claims rely on the spatial distinction between IFE and HFSCs, so tail skin is the most reliable tissue to address these issues.
2. In Figure 4 and subsequent analyses depicted in Figure 5, the gating strategy used to separate HFSCs from IFE SCs seems flawed. By sorting out Itga6^{high} cells that are CD34⁺, the isthmus and junctional zone hair follicle stem cells are not included. The authors should state that by sorting CD34⁺ cells, they enrich for bulge stem cells and not HFSCs. However, bulge stem cells have been shown to demonstrate a surprising plasticity and can migrate to the IFE in response to activation of the immune system, such as injury repair. Have the authors checked whether CD34⁺ stem cells migrate to the IFE in conditions where c-Jun and JUNB are depleted from keratinocytes? As many of the claims in this manuscript are based on this sorting strategy, this should be checked. Also, CD34 is expressed by a variety of immune and vascular cells, so this should be considered for flow cytometric analysis.
3. Related to the previous comment: In EV3A, the authors depict the gating strategy for quantification of different epidermal stem cell populations. There are several issues with these plots. CD34-APC staining seems to be very inefficient, as staining should result in a clear shift in fluorescence (see Jensen et al., 2010). As a major part of the manuscript relies on the isolation of these cell-types based on this gating strategy, this represents an issue that should be addressed.
4. The authors show a reduction of c-Jun and JUNB in human bulge cells of psoriatic patients. Are these genes downregulated in the K15⁺ stem cell population in inflammatory conditions? How about in the other populations of epidermal stem cells, such as Lgr5⁺, Gata6⁺, Gli1⁺ stem cells?
5. The authors claim that K15 staining in human skin is confined to the bulge of hair follicles. However, in Figure 1a, there is clear staining of K15 in the IFE.
6. TSLP production by mutant cells is investigated as a paracrine factor regulating the early events in psoriasiform disease. However, many other pro-inflammatory mediators, such as IL-6 and several chemokines are clearly downregulated in mutant HFSCs. This is somewhat surprising. In order to prove that TSLP production has paracrine effects, the authors should block TSLP in their co-culturing set-up.

Minor comments:

1. Hair follicle stem cells exhibit a striking plasticity in inflammatory conditions. The plasticity of K15⁺ stem cells, enabling these cells to migrate to the IFE in conditions of immune activation are not sufficiently mentioned in the introduction/discussion of the manuscript.
2. The authors show that IL-6 levels are highly increased in serum of mice at day 30 after induction of the phenotype. The authors claim this is a phenomenon that does not occur at earlier timepoints. In order to prove this claim, serum levels of IL-6 should be studied at earlier time-points of disease.
3. Several claims are overstated, such as the conclusion that DKO HF-SCs grow faster in culture than their wild-type counterparts, while there is no statistically significant difference between the growth curves.
4. How has cell death been analysed and quantified in in vitro settings?
5. Error bars and statistical analyses are lacking in EV6 panel I, J, K.
6. The authors are strongly advised to go through the manuscript and remove phrases like 'add ref.' et cetera.
7. Statistical significance is often not properly represented. For example: Fig4G: which conditions have been compared?
8. This reviewer would advise to change the title to 'Role of bulge epidermal stem cells and TSLP

signaling in psoriasis-like disease', as many inflammatory skin conditions, such as atopic dermatitis or wound healing

1st Revision - authors' response

11 July 2019

Point-to-point response to reviewers' comments

Referee #1 (Comments on Novelty/Model System for Author):

I believe the lineage tracing studies are well done and the in vitro studies indeed reveal intriguing means by which mutant and non mutant stem cells interact and drive psoriasis. This study also raises the possibility of targeting TSLP in psoriasis or stem cell derived factors in disease.

Referee #1 (Remarks for Author)

In their Article "Role of epidermal stem cell populations and TSLP signaling in inflammatory skin disease", Gago Lopez and colleagues examine the contribution of distinct epithelial stem and progenitor subsets to psoriasis like skin inflammation. The use tissue specific loss of cJun/JUNB to assess the contribution distinct progenitors to disease induction and inflammatory pathology. In doing so they uncovered that mutant cells lacking cJun/JUNB can influence the behavior of neighboring non-mutant cells, building upon their previously published idea that loss of cJun/JUN in epithelia is sufficient to induce inflammation. They identify TSLP as a secreted mediator responsible for communication between mutant and non-mutant cells. This is an interesting article and well suited for EMBO Molecular Medicine. I have the following concerns that should be addressed prior to publication.

We thank the reviewer for her/his positive and constructive evaluation.

1) For lineage tracing students in figure 2 and 3 in many panels the D0 time points are missing, it's important to have these data as they indicate the initial cells that are markers and may change between D0 and D5 (presented)

This is a very valid concern. Representative images at time 0 after induction have been included in Figure 2C, EV 2D and EV 3E. We observed groups of GFP⁺ mutant epidermal cells in the IFE and HF of DKO* mice, while DKO*¹⁵ mice expressed GFP⁺ epidermal cells mainly in the HFs and in some isolated epidermal cells of IFE as it was expected.

2) Figure 4B the authors make the conclusion that HFSCs are less prone to cell death, however information about confluence of cultures is not included or discussed. Are basal KCs which clearly proliferate at a higher rate reaching confluences sooner and thus also more prone to cells death. Along those lines it is not clear in 4F why co-culture pf GFP+ cells with DKO cells results in lower numbers than in 4E. Also does panel G use HF-SCs or KCs for their studies. These issues need to be clarified/addressed in the legends.

We thank the reviewer for her/his important suggestion. The culture conditions are described in Methodology (page 31-32) and we have added detailed information in the Results section on page 13-14. Briefly, both populations, bulge HF-SCs and basal-KCs were cultured in low concentration (2.000 cells per cm²) and three days later, when cell confluence was 25% in both populations, time-lapse images were captured from small colonies of keratinocytes during 48 hours. Regarding the comment: "lines it is not clear in 4F why co-culture pf GFP+ cells with DKO cells results in lower numbers than in 4E", we have mentioned in the Result section that "non-mutant^{Tom} bulge HF-SCs conditioned the proliferation of mutant^{GFP} bulge HF-SCs, whereas non-mutant^{Tom} b-KCs did not affect the proliferation of mutant^{GFP} b-KCs (Fig 4 E, F)". Finally, the trans-well assay in Figure 4G has been done with basal KCs which is mentioned in the Results on page 15.

3) It is unclear why the authors choose to show Ki67 in 6L and quantify Edu cells in 6K. Since Edu is a more faithful marker of proliferating cells, this should be used but for image analysis and quantification.

This is a valid concern: In Figure 6L, we quantified proliferative epidermal cell in a pulse-chase analysis with EDU (2 hours) by FACS analysis. We used the proliferation marker Ki67 to identify the growth region that includes reduction of epidermal hyperplasia. Therefore, we used two different techniques to confirm our data. We have rephrased this part on page 21 as follow: "Proliferative epidermal cells were significantly reduced in both mouse models after anti-TSLP treatment (Fig.

6K), confirmed by a reduction of epidermal hyperplasia and proliferation by Ki67 expression in ear sections (Fig. 6L)

4) Is TSLP sufficient to induce proliferation in tomato+ KCs? KCs from WT mice? In other words, are other factors from the GFP+ cell environment priming the non-mutant cells? Along those lines the authors assay for TSLP α but not IL7 α which is also required for TSLP signaling.

This is an important point to improve the mechanistic aspect of our manuscript. In order to demonstrate that WT KCs can respond to TSLP, we have performed a proliferation assay in primary KCs treated with recombinant TSLP *in vitro* (Figure EV 6 A-C). Primary WT KCs increased proliferation rate after recombinant TSLP treatment, even at low concentration of recombinant TSLP. We also performed the treatment with recombinant TSLP in primary WT KCs derived from bulge HF-SCs (CD34+/CD49f+) and basal KCs (CD34-/CD49f+). Interestingly, bulge HF-SCs derived KCs significantly increased the proliferation rate in comparison with basal KCs (see page 19).

We have also performed qPCR to analyse gene expression of IL7 α in the *in vitro* TSLP neutralization experiment (Figure 6E). We observed that IL-7 α expression increased after the neutralization of TSLP in co-culture (Fig. 6E), while the specific receptor for TSLP (TSLPR) decreased as expected (Fig. 6D). We hypothesize that IL-7 α expression increased after TSLP blocking as part of a compensatory mechanism in response to TSLP neutralization (page 19).

5) In general, the legends could use more clarity and details. The figures are often hard to understand based on the results and legends.

We thank the Reviewer for her/his suggestion and have rephrased the legends to clarify these issues.

Referee #2 (Comments on Novelty/Model System for Author):

1. The methodology is appropriate and the data shown convincing.
2. The authors indicate that the decreased expression of c-Jun and JunB in hair follicle stem cells is sufficient for psoriasis-like disease. This is quite novel.
3. Psoriasis is a common inflammatory disease in the skin. Understanding the mechanisms of psoriasis development will help the treatment of patients.
4. The authors used model mice adequately.

Referee #2 (Remarks for Author):

Psoriasis is a common inflammatory disease in the skin. Understanding the mechanisms of psoriasis development will help the treatment of patients.

In this article, Gago-Lopez et al found the decreased expression of c-JUN and JUNB in the bulge region of hair follicles, known as a hair follicle stem cell niche, in psoriatic patients. The authors then confirmed that decreased expression of c-Jun/JunB in hair follicle stem cells is sufficient for the initiation of psoriasis-like disease using mutant mice previously generated by the authors. The authors also identified TSLP as a paracrine factor that is secreted from mutant hair follicle stem cells and mutant basal keratinocytes. TSLP induces proliferation of neighboring non-mutant keratinocytes and VEGF expression, which results in the development and maintenance of a psoriasis-like inflammatory skin disease.

This is an interesting study with clear medical relevance. The methodology is appropriate and the data shown convincing. Altogether, I recommend publication of this manuscript in EMBO Molecular Medicine pending the authors address the following comments.

We thank the Reviewer for her/his positive and constructive evaluation.

Major comments:

This manuscript starts with the human clinical samples where c-JUN and JUNB are decreased in the bulge region of hair follicles in psoriatic patients (Fig. 1). And the latter of this study, the authors identify TSLP as a mediator of psoriasis-like disease in c-Jun/JunB mutant mice. As the mouse model study indicated, is TSLP expression increased in the bulge region of hair follicles in psoriatic patients? The authors should confirm the expression of TSLP in psoriatic patients and indicate the results in the manuscript.

We have analysed TSLP expression in human scalp psoriatic patients (new Figure 7), and TSLP increased in lesional scalp of three independent samples of psoriatic patients. Interestingly, we observed constitutive expression of TSLP in the outer root sheath (ORS) of non-lesional (NL) hair

follicles (Figure 7A, 7D) and/or healthy hair follicles (data not shown) mainly in the sub-bulge region (Figure 7D, 7E). In psoriatic patients, the expression of TSLP was significantly increased in the scalp epidermis and in the ORS of the sub-bulge region of HFs (Figure 7A-C, D, E), whereas TSLP was not expressed in the bulge region of HFs labelled with CD200 from psoriatic patients (non-lesional and lesional scalp) (Figure 7E and page. 22). This text was also added to the Discussion on page 26.

In Fig. 2I, the authors describe that mutant GFP⁺ keratinocytes are maintained around hair follicles. But it is very hard to confirm the location of GFP-positive cells. The authors should also indicate the histological sections.

We have changed the image of Figure 2H to a more clear image that shows expression of GFP-positive cells in hair follicles.

In Fig. 4B, the authors indicate that HF-SC-derived mutant GFP⁺ KCs show long-term proliferative capacity and mutant GFP⁺ b-KCs show only short-term proliferative capacity. But, the stem cell population is enriched in the culture of HF-SC population-derived KCs. In contrast, the culture of b-KCs consists of keratinocytes with a variety of growth potentials. This experiment simply shows the difference of proliferative capacity between stem cell and transient amplifying cell populations. Therefore, the authors cannot claim that HF-SC-derived mutant KCs has different properties in proliferation and survival using this experimental model.

This is a very valid concern raised by the Reviewer. In Figure 4B and 4C we analysed the behaviour of mutant GFP⁺ keratinocytes derived from HF-SCs and basal KCs (green line in the graph) and compared to their control respective populations (WT HF-SCs and WT basal KCs, grey line in the graph). All cells were cultured in serum-free keratinocyte media (KGM) to reduce the effect of external signalling (see Methodology on page 31-32). WT basal KCs grow similarly than WT keratinocytes derived from HF-SCs (grey line in the graphs 4B, 4C). However, while mutant GFP⁺ basal KCs acquired a high proliferative rate during the first 24 hours of time lapse capture, they underwent cell death in the next 24 hours of time lapse capture, when compared to WT control basal KCs. In contrast, mutant GFP⁺ KCs from HF-SCs maintained a similar growth to control WT HF-SCs KCs during 48 hours of capture without an increased in cell death. We have rephrased the text in Results on page 13-14 and Figure legend to clarify these data.

In Fig. 5C, the authors indicate the 4 genes coding for secreted proteins and then investigated TSLP in detail. Are other 3 genes not involved in the development of psoriasis? Otherwise, the authors should simply mention and/or discuss about that.

We have mentioned the 3 genes in results in page 17-18 and their relationship with inflammatory skin diseases.

In Fig. 6H, the authors clearly demonstrate that anti-TSLP inhibits psoriasis-like disease development. Does TSLP neutralization also alleviate psoriasis-like disease after the development of this disease? The authors indicate that mutant b-KCs are eliminated in the late -stage but mutant HF-SCs are maintained and contribute to the chronic skin inflammation. But the authors did not show any evidence that mutant HF-SC contributes to the maintenance of this disease.

The experiment *in vivo* shown in Fig. 6H was performed during the initial phase-of psoriasis-like development (Figure 6G), and therefore the ears are already affected (see ear thickness and TEWL at day 0 of treatment in Figure 6 I,J).

Regarding the contribution of mutant HF-SCs for the maintenance of the disease, we suggest that mutant HF-SCs may act as a reservoir of mutant KCs to maintain the chronicity over time since mutant basal KCs are eliminated and activated mutant bulge HF-SCs give rise new mutant KCs that migrate to the epidermis and maintain the disease. However, as the reviewer suggested, we demonstrated that mutant bulge HF-SCs initiate psoriasis development but we do not have additional data to confirm that these stem cells also maintain the disease. We have removed “maintain” in the text on page 11 to avoid confusion.

Minor comments

In Page 8, line 1-3, the authors describe the induction of GFP with tamoxifen treatment. This result is indicated in EV 1A, right panels, but not EV 1B, upper panels.

We wrote: “*We first analyzed the labeling efficiency and epidermal specificity of GFP in Co-mT/mG ear skin 15 days after tamoxifen injection. Without tamoxifen, 95±5% of epidermal cells*

expressed Tomato and only few keratinocytes expressed GFP (EV 1A). Following tamoxifen treatment, Tomato expression decreased and GFP expression increased by ~60-70% in K5⁺ epidermal cells of IFE, hair follicles and sebaceous glands (EV 1B, upper panels)". We add EV1B, upper panels in the text in page 8.

In Page 17, lines 2-3, the authors describe that "displayed a 12-fold increase in mutantGFP HF-SCs and a 4-fold increase ...". But in Fig. 5C, the fold change in HS-SCGFP is 6.17 and that of b-KCGFP is 2.75. how do readers interpret this?

Figure 5C represents fold change in Log2; we have modified these data in Figure 5C.

Referee #3 (Remarks for Author):

In the manuscript 'Role of epidermal stem cell populations and TSLP signaling in inflammatory skin disease' by Gago-Lopez et al., the authors claim that hair follicle stem cells are sufficient to initiate psoriasis-like disease in murine skin. Their major findings are: 1) levels of c-Jun and JUNB are reduced in hair follicle stem cells from human psoriatic patients; 2) Lineage tracing studies revealed that IFE keratinocytes that are depleted of c-Jun and JUNB exhibit different proliferation and apoptosis rates in psoriasis-like disease; 3) Deletion of c-Jun and JUNB specifically in the K15⁺ population of HFSCs results in the development of a psoriasis-like phenotype that is milder as the inflammatory phenotype observed in mice lacking c-Jun and JUNB in all keratinocytes; 4) secreted factors from keratinocytes that lack c-Jun and JUNB induce hyperproliferation in wild-type neighboring epidermal cells and molecular signatures differ between both cell types; 5) secretion of TSLP by keratinocytes that lack c-Jun and JUNB induces hyperproliferation of keratinocytes and blocking TSLP in vivo and in vitro ameliorates the inflammatory features that are linked to deletion of c-Jun and JUNB from HFSCs.

This is an interesting study showing a role for c-Jun and JUNB specifically in HFSCs, however there are various concerns as several claims are not properly backed up by experimental evidence.

We thank the Reviewer for her/his positive and constructive evaluation.

Major concerns:

1. All in vivo studies have been conducted on ear skin. Hair follicles in hyperplastic ear skin are often not clearly distinguishable from the IFE (see Figure 2C, H and I; Figure 3). These analyses should be repeated on tail skin, where the different stem cell populations are spatially restricted and the difference between IFE and HFSCs is much more obvious. Many of the major claims rely on the spatial distinction between IFE and HFSCs, so tail skin is the most reliable tissue to address these issues.

This is a valid concern. Our GEMM for psoriasis-like disease develops psoriatic-like plaques mainly in ear skin, tail and paws. While ear skin is always affected by psoriasis-like disease, there is some variability in the psoriatic-like plaque formation in the tail. In order to have reproducible experiments, we chose ear skin for all our study. However, we understand the point raised by the Reviewer and we have carried out histological sections of tail skin from animals with psoriasis-like phenotype and add these images in the new Appendix Figure S2-A. We analysed the expression of GFP during psoriasis-like progression (Day 5-Day 45) in the tail skin of psoriatic DKO* mice. We observed the same dynamic loss of epidermal GFP⁺ expression in the IFE, whereas HF⁺ maintained GFP expression along psoriasis-like progression. As we showed in ear, we also observed long hair follicles in anagen phase along psoriasis-like progression which suggest the activation of the hair follicle growth cycle by mutant GFP⁺ HF-SCs. We also shown histological section of back skin of psoriatic DKO* mice with similar pattern than in ear or tail skin (Appendix Fig S2-B). These data have been added to the Result section on page 10.

2. In Figure 4 and subsequent analyses depicted in Figure 5, the gating strategy used to separate HFSCs from IFE SCs seems flawed. By sorting out Itga6^{high} cells that are CD34⁺, the isthmus and junctional zone hair follicle stem cells are not included. The authors should state that by sorting CD34⁺ cells, they enrich for bulge stem cells and not HFSCs. However, bulge stem cells have been shown to demonstrate a surprising plasticity and can migrate to the IFE in response to activation of the immune system, such as injury repair. Have the authors checked whether CD34⁺ stem cells migrate to the IFE in conditions where c-Jun and JUNB are depleted from keratinocytes? As many of the claims in this manuscript are based on this sorting strategy, this should be checked. Also,

CD34 is expressed by a variety of immune and vascular cells, so this should be considered for flow cytometric analysis.

We have added “bulge” in the text of figure/legends to clarify the origin of CD34+ HF-SC population. In addition, we have analyzed the expression of CD34 in the IFE of psoriatic mice and we did not find CD34+ cells in the IFE of DKO* psoriatic mice, only in the bulge of hair follicles (Appendix Figure S4-A) explained now in Page 11 of results. Regarding FACS analysis and sorting for CD34+ HF-SC populations, we discarded CD34 cells positive for CD45 (hematopoietic cells) and CD31 (endothelial cells) in our analysis. Gating strategy is explained now in Appendix Figure S4-B and on page 11.

3. Related to the previous comment: In EV3A, the authors depict the gating strategy for quantification of different epidermal stem cell populations. There are several issues with these plots. CD34-APC staining seems to be very inefficient, as staining should result in a clear shift in fluorescence (see Jensen et al., 2010). As a major part of the manuscript relies on the isolation of these cell-types based on this gating strategy, this represents an issue that should be addressed.

We have noted that the CD34 expression in the bulge region of ear hair follicles is lower than in tail or back skin. Therefore, the intensity of fluorescence by FACS is also lower, although sufficient enough to distinguish bulge CD34+ HF-SCs from *Inta6+* basal KCs (see Appendix Figure S4-B).

4. The authors show a reduction of c-Jun and JUNB in human bulge cells of psoriatic patients. Are these genes downregulated in the K15+ stem cell population in inflammatory conditions? How about in the other populations of epidermal stem cells, such as *Lgr5+*, *Gata6+*, *Gli1+* stem cells? In Figure 1 we analysed c-JUN and JUNB expression in the K15+ stem cell population from healthy patients and psoriatic patients and we observed a reduction in c-JUN and JUNB expression into this K15+ stem cell population in the bulge (Figure 1 A-J). We have added K15 in the graphs of Fig. 1I and J to clarify this point.

While *Lgr5*, *Gata-6* and *Gli1* are described to identify stem cell subpopulations into mouse hair follicles, it is unclear if these markers label the same subpopulations in human hair follicles. However, other markers such as CD200 identifies human HF-SCs in the bulge region of human HFs. We analysed the expression of CD200 together c-JUN and JUNB in non-lesional and lesional psoriatic scalp, and we observed that these specific bulge HF-SCs downregulated both transcription factors (EV 1 A). GATA-6 has been described to be expressed in the matrix region (bulb) of mouse hair follicles but the expression in human hair follicles is unclear. We have analysed the expression of GATA-6 in human scalp from psoriatic patients (EV 1B,C). Some cells from the basal layer of the ORS in the proximal bulb (Pb) region expressed GATA-6 in both samples, non-lesional and lesional scalp. In addition, these cells co-expressed c-JUN and JUNB. Interestingly, in the bulge region where c-JUN and JUNB are downregulated in human psoriatic bulge HF-SCs, we observed overexpression of GATA-6 in adjacent suprabasal layer cells in the ORS of lesional HFs vs non-lesional samples.

In conclusion, we have extended the characterization of epidermal stem cell subpopulations in human psoriatic samples and the co-expression of c-JUN and JUNB in psoriasis on page 6-7.

5. The authors claim that K15 staining in human skin is confined to the bulge of hair follicles. However, in Figure 1a, there is clear staining of K15 in the IFE.

We agree with the Reviewer. In human scalp, K15 is highly expressed in the bulge of hair follicles, but low expression also exists in some basal keratinocytes from IFE. We have clarified this point on page 6.

6. TSLP production by mutant cells is investigated as a paracrine factor regulating the early events in psoriasis disease. However, many other pro-inflammatory mediators, such as IL-6 and several chemokines are clearly downregulated in mutant HFSCs. This is somewhat surprising. In order to prove that TSLP production has paracrine effects, the authors should block TSLP in their co-culturing set-up.

In this mouse model for psoriasis, IL-6 is upregulated in non-mutant^{Tomato} HF-SCs and basal KCs (EV 4J) while the IL-6ra is overexpressed in mutant^{GFP} HF-SCs (by RNAseq data). Mutant^{GFP} HF-SCs upregulated other types of pro-inflammatory mediators such as *Tnf-α* or *Ptgs2* (EV 4 F, G), also important mediators in psoriasis. In order to clarify the autocrine and paracrine effect of TSLP, we neutralized TSLP in the co-culture system described in Figure 6A. Then, we purified mutant^{GFP} and non-mutant^{Tomato} KCs by FACS sorting to determine the gene expression for different pro-inflammatory mediators in psoriasis (EV 5 D-M). We confirmed that VEGF-α is overexpressed in

both populations, mutant^{GFP} and non-mutant^{Tom} KCs after induction with adeno CRE, while its expression was downregulated after TSLP neutralization as we initially showed in the Fig. 6F. IL-6 was upregulated in non-mutant^{Tom} KCs, while was downregulated after TSLP blocking. Interestingly, other pro-inflammatory cytokines and mediators such as IL-1 β , and p65 were upregulated in mutant^{GFP} KCs after the deletion of cJun/JunB and their expression were downregulated after TSLP neutralization. Furthermore, IFN- γ or G-CSF were upregulated in both populations, mutant^{GFP} and non-mutant^{Tom} KCs, however only mutant^{GFP} KCs responded to TSLP neutralization.

Overall, we suggest that TSLP acts as an autocrine and paracrine factor in the cross-talk between mutant and non-mutant KCs and regulates the expression of different pro-inflammatory mediators (see Results on page 20).

Minor comments:

1. Hair follicle stem cells exhibit a striking plasticity in inflammatory conditions. The plasticity of K15+ stem cells, enabling these cells to migrate to the IFE in conditions of immune activation are not sufficiently mentioned in the introduction/discussion of the manuscript.

We have clarified the plasticity of K15+ in the migration of IFE in other diseases including references (please see page 3-4).

2. The authors show that IL-6 levels are highly increased in serum of mice at day 30 after induction of the phenotype. The authors claim this is a phenomenon that does not occur at earlier timepoints. In order to prove this claim, serum levels of IL-6 should be studied at earlier time-points of disease. We have double-checked this statement, and we do not mention that IL-6 is not induced at earlier timepoints.

3. Several claims are overstated, such as the conclusion that DKO HF-SCs grow faster in culture than their wild-type counterparts, while there is no statistically significant difference between the growth curves.

We agree with the Reviewer and have modified this statement on page 14.

4. How has cell death been analysed and quantified in in vitro settings?

Cell death was analysed by Topro3 and DAPI. We have added this information in Methodology on page 32.

5. Error bars and statistical analyses are lacking in EV6 panel I, J, K.

These data were added.

6. The authors are strongly advised to go through the manuscript and remove phrases like 'add ref.' et cetera.

We have revised the manuscript and made sure that these phrases are corrected/omitted.

7. Statistical significance is often not properly represented. For example: Fig4G: which conditions have been compared?

This was clarified in Figure legends.

8. This reviewer would advise to change the title to 'Role of bulge epidermal stem cells and TSLP signaling in psoriasis-like disease', as many inflammatory skin conditions, such as atopic dermatitis or wound healing

Many thanks for the suggestion, which we will accept for the title.

2nd Editorial Decision

9 August 2019

Thank you for the submission of your revised manuscript to EMBO Molecular Medicine, and please accept my apologies for the unusual delay in getting back to you, which is due to the fact that one referee needed more time to complete his/her evaluation. We have now received both referees' reports, and as you will see the referees are now supportive of publication of your work. I am therefore pleased to inform you that we will be able to accept your manuscript pending the following final editorial amendments.

***** Reviewer's comments *****

Referee #2 (Remarks for Author):

The authors addressed all remarks and the manuscript is now suitable for publication in EMBO molecular medicine.

Referee #3 (Remarks for Author):

The authors have answered my concerns in an adequate manner and I would advise for publication of the revised manuscript.

2nd Revision - authors' response

27 August 2019

The authors addressed all minor editorial concerns.

Corresponding Author Name: Erwin F. Wagner
Journal Submitted to: EMBO Molecular Medicine
Manuscript Number: EMM-2019-10697